# Psychological characteristics associated with COVID-19 vaccine hesitancy and resistance in Ireland and the United Kingdom

Jamie Murphy [1✉], Frédérique Vallières [2], Richard P. Bentall [3], Mark Shevlin [1], Orla McBride[1], Todd K. Hartman [3], Ryan McKay [4], Kate Bennett[5], Liam Mason[6], Jilly Gibson-Miller [3], Liat Levita [3], Anton P. Martinez[3], Thomas V. A. Stocks [3], Thanos Karatzias [7] & Philip Hyland [8]

Identifying and understanding COVID-19 vaccine hesitancy within distinct populations may aid future public health messaging. Using nationally representative data from the general adult populations of Ireland ($N = 1041$) and the United Kingdom (UK; $N = 2025$), we found that vaccine hesitancy/resistance was evident for 35% and 31% of these populations respectively. Vaccine hesitant/resistant respondents in Ireland and the UK differed on a number of sociodemographic and health-related variables but were similar across a broad array of psychological constructs. In both populations, those resistant to a COVID-19 vaccine were less likely to obtain information about the pandemic from traditional and authoritative sources and had similar levels of mistrust in these sources compared to vaccine accepting respondents. Given the geographical proximity and socio-economic similarity of the populations studied, it is not possible to generalize findings to other populations, however, the methodology employed here may be useful to those wishing to understand COVID-19 vaccine hesitancy elsewhere.

[1] School of Psychology, Ulster University, Coleraine BT52 1SA, Northern Ireland. [2] Centre for Global Health, Trinity College Dublin, Dublin D02 PN40, Republic of Ireland. [3] Department of Psychology, University of Sheffield, Sheffield S10 2TN, England. [4] Department of Psychology, Royal Holloway, University of London, London TW20 0EX, England. [5] Department of Psychology, University of Liverpool, Liverpool L69 3BX, England. [6] Division of Psychology and Language Sciences, University College London, London WC1E 6BT, England. [7] School of Health and Social Care, Napier University, Edinburgh EH14 1DJ, Scotland. [8] Department of Psychology, Maynooth University, County Kildare W23 F2K8, Republic of Ireland. ✉email: ja.murphy@ulster.ac.uk

Severe acute respiratory syndrome coronavirus 2 (SARS-CoV-2), the virus that causes COVID-19, reached pandemic status on March 11th, 2020. As of September 11th, 2020, the virus had spread to 213 countries and territories, infected over 28 million people, and resulted in over 900,000 deaths worldwide[1]. The global economic cost of this pandemic over the next 2 years is projected to lead to a cumulative output loss of nine trillion US dollars[2]. In the absence of an effective therapy or vaccine, governments around the world enacted extreme physical distancing and quarantine measures to slow the spread of the virus, protect the most vulnerable in society, and manage health care service demand and provision[3]. The necessity for an approved vaccine to protect populations from this virus, as well as to safeguard economies from continued disruption and damage, cannot be overstated.

The first human clinical trial of a COVID-19 vaccine commenced on March 3rd, 2020 in the United States[4], and several other human trials commenced soon after[5]. As of September 11th, 2020, 8 vaccines had advanced to Phase 3 clinical trials and 2 had been approved for early or limited use[6]. Many trials are ongoing. In an April 2020 study of 7664 people drawn from seven European nations (Denmark, France, Germany, Italy, Portugal, the Netherlands, and the United Kingdom (UK)), 18.9% of respondents indicated that they were 'unsure' about taking a vaccine for COVID-19, while a further 7.2% indicated that they did not want to get vaccinated[7]. Identifying, understanding, and addressing vaccine acceptance (i.e. a position ranging from passive acceptance to active demand)[8], and vaccine hesitance and resistance (i.e. the positions where one is unsure about taking a vaccine or where one is absolutely against taking a vaccine)[9] to a vaccine for COVID-19 is, therefore, a potentially important step to ensure the rapid and requisite uptake of an eventual vaccine.

Much of the existing literature on vaccine hesitance and resistance focuses on the explicit reasons individuals provide for their opposition to a particular vaccine or to vaccination programmes in general[9–12]. Although useful, this information is limited in terms of its ability to explain why individuals come to their respective epistemological positions[13]. A more informative approach may be to identify the psychological processes that characterise and distinguish vaccine hesitant and resistant individuals from those who are receptive to vaccines; an approach that is reflective of the "attitude roots" model of science rejection[14]. Doing so not only helps to account for why vaccine hesitant and resistant individuals come to hold the specific beliefs that they do, but it may also provide an opportunity to tailor public health messages in ways that are consistent with these individuals' psychological dispositions. Given that public service campaigns advocating a variety of health behaviours have benefitted from psychologically oriented approaches[15–17], public health messaging efforts aimed at increasing the uptake of a COVID-19 vaccine can benefit from a comprehensive understanding of the psychology of vaccine hesitant and resistant individuals.

To date, a number of psychological constructs have been explored in relation to vaccine hesitancy. For example, altruistic beliefs[18], the personality traits neuroticism and conscientiousness[19,20], locus of control[21], and cognitive reflection[22] have each been shown, in some way, to influence vaccine acceptance/hesitancy. Vaccine hesitance/resistance has also been associated with conspiratorial, religious, and paranoid beliefs[13,23–25], while mistrust of authoritative members of society, such as government officials, scientists, and health care professionals, has been linked to negative attitudes towards vaccinations[26–30], as has endorsement of authoritarian political views, societal disaffection, and intolerance of migrants[31,32]. Taken together, the existing literature indicates that there are likely to be several psychological dispositions that traverse personality, cognitive styles, emotion, beliefs, trust, and socio-political attitudes that distinguish those who are hesitant or resistant to a COVID-19 vaccine from those who are accepting.

With emerging research findings indicating that a substantial proportion of European adults are hesitant about, or resistant to, a vaccine for COVID-19[7], important work is required to begin to understand and address this problem. The importance of identifying, describing, and understanding these individuals as a key preparatory step for vaccine development is further emphasised by the World Health Organization's (WHO, 2014) Strategic Advisory Group of Experts (SAGE) on Immunisation[33]. It is imperative, therefore, that we begin to understand the psychological characteristics that define and distinguish those who are hesitant and resistant to a vaccine for COVID-19 from those who are accepting. To achieve these goals, we developed four study objectives.

First, we sought to determine what proportions of the general adult populations of Ireland and the UK were accepting of, hesitant about, or resistant to a vaccine for COVID-19.

Second, we sought to profile individuals who are hesitant about, or resistant to, a possible vaccine for COVID-19 by identifying the key sociodemographic, political, and health-related factors that distinguish these individuals from those who are accepting of a COVID-19 vaccine. By identifying these distinguishing, objective characteristics, public health officials may be better able to identify who in the population is more likely to be hesitant or resistant to a COVID-19 vaccine.

Third, we sought to identify the most salient psychological characteristics that distinguish individuals who are hesitant/resistant to a COVID-19 vaccine from those who are accepting. A better understanding of the psychology of vaccine hesitant and resistant individuals affords public health officials a more complete understanding of why these individuals view a COVID-19 vaccine the way that they do.

Finally, we sought to determine from which sources vaccine hesitant and resistant individuals gather information about the COVID-19 pandemic, as well as the level of trust they place in these sources. Taken together, these latter two objectives offer a greater understanding of how public health officials can effectively tailor health behaviour messaging to align to the psychological profiles of vaccine hesitant or resistant individuals, while also taking into account their consumption and trust proclivities relating to COVID-19 information.

## Results
Data from nationally representative samples of the general adult populations of Ireland ($N = 1041$) and the UK ($N = 2025$) were collected. The sociodemographic characteristics for both samples are reported in Table 1.

**Objective 1: prevalence of vaccine hesitancy and resistance in Ireland and the UK.** Overall, 65% (95% CI = 62.0, 67.9) of Irish respondents were accepting of a COVID-19 vaccine, 26% (95% CI = 22.9, 28.3) were hesitant about such a vaccine, and 9% (95% CI = 7.7, 11.3) were resistant to such a vaccine. Comparatively, 69% (95% CI = 66.8, 70.9) of UK respondents were vaccine accepting, 25% (95% CI = 23.1, 26.9) were vaccine hesitant, and 6% (95% CI = 5.2, 7.3) were vaccine resistant. Figure 1 displays the proportions in these three groups for Ireland and the UK overall, as well as for its devolved nations of England, Wales, Scotland, and Northern Ireland. As can be seen, Northern Ireland had the lowest rate of vaccine acceptance at 51%.

**Objective 2: sociodemographic, political, and health variables associated with COVID-19 vaccine hesitancy and resistance.**

**Table 1 Sociodemographic characteristics of the Irish and UK samples.**

| Ireland (N = 1041) | % | UK (N = 2025) | % |
|---|---|---|---|
| *Sex* | | *Sex* | |
| Female | 51.5 | Female | 51.7 |
| Male | 48.2 | Male | 48.3 |
| *Age* | | *Age* | |
| 18–24 | 11.1 | 18–24 | 12.1 |
| 25–34 | 19.2 | 25–34 | 18.8 |
| 35–44 | 20.6 | 35–44 | 17.4 |
| 45–54 | 15.9 | 45–54 | 20.2 |
| 55–64 | 21.0 | 55–64 | 17.2 |
| 65+ | 12.2 | 65+ | 14.2 |
| *Birthplace* | | *Birthplace* | |
| Ireland | 70.7 | UK | 90.6 |
| *Region of Ireland* | | *Region of UK* | |
| Leinster | 55.3 | England | 86.9 |
| Munster | 27.3 | Scotland | 7.8 |
| Connaught | 12.0 | Wales | 3.1 |
| Ulster | 5.4 | Northern Ireland | 2.3 |
| *Ethnicity* | | *Ethnicity* | |
| Irish | 74.8 | White British/Irish | 85.5 |
| Irish Traveller | 0.3 | White non-British/Irish | 5.7 |
| Other White background | 17.3 | Indian | 2.0 |
| African | 1.9 | Pakistani | 1.3 |
| Other Black background | 0.3 | Chinese | 0.9 |
| Chinese | 0.4 | Afro-Caribbean | 0.6 |
| Other Asian | 3.2 | African | 1.3 |
| Mixed background | 1.8 | Arab | 0.1 |
| | | Bangladeshi | 0.3 |
| | | Other Asian | 0.5 |
| *Living location* | | *Living location* | |
| City | 24.5 | City | 24.6 |
| Suburb | 18.1 | Suburb | 28.2 |
| Town | 26.8 | Town | 30.6 |
| Rural | 28.8 | Rural | 16.5 |
| *Highest education* | | *Highest education* | |
| No qualification | 1.2 | No qualifications | 2.9 |
| Finished mandatory schooling | 6.4 | O-level/GCSE or similar | 19.0 |
| Finished secondary school | 22.4 | A-level or similar | 18.1 |
| Undergraduate degree | 22.5 | Diploma | 5.6 |
| Postgraduate degree | 19.8 | Undergraduate degree | 28.2 |
| Other technical qualification | 27.9 | Postgraduate degree | 15.6 |
| | | Technical qualification | 9.3 |
| | | Other | 1.3 |
| *2019 income* | | *2019 income* | |
| 0–€19,999 | 24.6 | £0–£15490 | 20.2 |
| €20,000–€29,999 | 21.3 | £15,491–£25,340 | 20.2 |
| €30,000–€39,999 | 19.5 | £25,341–£38,740 | 19.0 |
| €40,000–€49,999 | 12.7 | £38,741–£57,930 | 20.2 |
| €50,000+ | 21.9 | £57,931+ | 20.2 |
| *Employment status* | | *Employment status* | |
| Full-time (self)/employed | 43.3 | Full-time (self)/employed | 48.8 |
| Part-time (self)/employed | 15.7 | Part-time (self)/employed | 15.0 |
| Retired | 15.0 | Retired | 16.5 |
| Unemployed | 8.4 | Unemployed | 11.7 |
| Student | 6.3 | Student | 4.7 |
| Unemployed (disability or illness) | 5.6 | Unemployed (disability or illness) | 3.4 |
| Unemployed due to COVID-19 | 5.7 | | |
| *Religion* | | *Religion* | |
| Christian | 69.8 | Christian | 50.4 |
| Muslim | 1.6 | Muslim | 3.0 |
| Jewish | 0.2 | Jewish | 0.8 |
| Hindu | 1.1 | Hindu | 0.6 |
| Buddhist | 0.6 | Buddhist | 0.8 |
| Sikh | 0.1 | Sikh | 0.5 |
| Other religion | 3.8 | Other | 6.0 |
| Atheist | 15.3 | Atheist | 25.4 |
| Agnostic | 7.5 | Agnostic | 12.5 |
| *Lone adult in household* | | *Lone adult in household* | |
| Yes | 18.4 | Yes | 22.4 |
| *Children in the household* | | *Children in the household* | |
| Yes | 39.7 | Yes | 29.2 |

The full set of findings from the multinomial logistic regression analyses for the Irish and UK samples are presented in Tables 2 and 3, respectively.

In the Irish sample, those who were vaccine hesitant – compared to those who were vaccine accepting – were more likely to be female (AOR = 1.62, 95% CI = 1.18, 2.22), aged between 35 and 44 years (AOR = 2.00, 95% CI = 1.06, 3.75), and less likely to have received treatment for a mental health problem (AOR = 0.63, 95% CI = 0.45, 0.88). Those who were vaccine resistant – compared to those who were vaccine accepting – were more likely to be aged 35–44 years (AOR = 3.33, 95% CI = 1.17, 9.47), residing in a city (AOR = 1.90, 95% CI = 1.02, 3.54), to be of non-Irish ethnicity (AOR = 2.89, 95% CI = 1.17, 7.09), to have voted for the political party Sinn Féin (AOR = 3.22, 95% CI = 1.14, 9.08) or an Independent political candidate (AOR = 4.15, 95% CI = 1.19, 14.49) in the previous general election, and to have an underlying health condition (AOR = 2.59, 95% CI = 1.38, 4.85). Income level was also associated with vaccine resistance where those in lower income brackets were more likely to be vaccine resistant.

Three variables distinguished those who were vaccine resistant from those who were vaccine hesitant: non-Irish ethnicity (AOR = 2.76, 95% CI = 1.05, 7.19), having an underlying health condition (AOR = 2.68, 95% CI = 1.33, 5.38), and having a lower level of income (AORs ranged from 2.82 to 5.44, 95% CIs ranged from = 1.04, 7.66 to 1.98, 14.93).

In the UK sample, those who were vaccine hesitant – compared to those who were vaccine accepting – were more likely to be female (OR = 1.43, 95% CI = 1.14, 1.80), and to be younger than 65. Those who were vaccine resistant – compared to those who were vaccine accepting – were more likely to be in younger age categories (over ten times more likely to be in the three lowest age categories, and over four times more likely to be aged 45–54 years or 55–64 years, than to be in the 65 and older category). They were also more likely to reside in a suburb (OR = 2.13, 95% CI = 1.01, 4.49), to be in the three lowest income brackets, and to be pregnant (OR = 2.36, 95% CI = 1.03, 5.40).

The only variable to distinguish vaccine resistant respondents from vaccine hesitant respondents in the UK sample was age. Those who were vaccine resistant were more likely to be in younger age categories (over seven times more likely to be aged 18–24, and over four times more likely to be aged between 25–34 years and 35–44 years, than to be in the 65 and older age category).

**Objective 3: psychological indicators of vaccine acceptance/ hesitancy/resistance.** The variation in measures of respondent psychology across the vaccine acceptance, hesitance, and resistance groups in the Irish and UK samples is reported in Tables 4 and 5, respectively.

In the Irish sample, the combined vaccine hesitant and resistant group differed most pronouncedly from the vaccine acceptance group on the following psychological variables: lower levels of trust in scientists ($d = 0.51$), health care professionals ($d = 0.45$), and the state ($d = 0.31$); more negative attitudes toward migrants ($d$'s ranged from 0.27 to 0.29); lower cognitive reflection ($d = 0.25$); lower levels of altruism ($d$'s ranged from 0.17 to 0.24); higher levels of social dominance ($d = 0.22$) and authoritarianism ($d = 0.14$); higher levels of conspiratorial ($d = 0.21$) and religious ($d = 0.20$) beliefs; lower levels of the personality trait agreeableness ($d = 0.15$); and higher levels of internal locus of control ($d = 0.14$).

When comparing the three groups in the Irish sample, the vaccine resistant group differed from the vaccine hesitant group in terms of higher levels of conspiracy beliefs ($\eta^2 = 0.02$), and lower levels of trust in scientists ($\eta^2 = 0.06$), health care professionals ($\eta^2 = 0.05$), and the state ($\eta^2 = 0.03$).

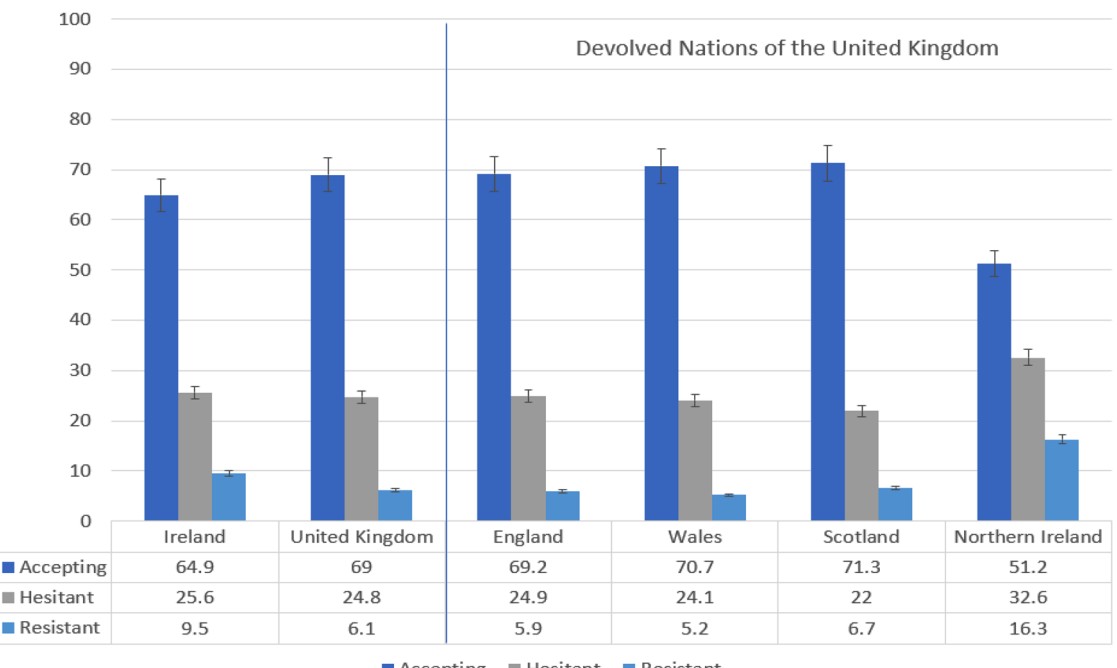

**Fig. 1 Rates of COVID-19 vaccine acceptance, hesitance, and resistance in Ireland and the United Kingdom (UK), and in the four devolved nations of the UK.** Data are presented as the proportion of the Irish ($N = 1041$) and United Kingdom ($N = 2025$) samples indicating COVID-19 vaccine acceptance (dark blue), hesitance (grey), and resistance (light blue) in the first two bar-charts on the left side of the figure. Error bars present the 95% confidence intervals of these proportions. The same information is presented for the four devolved nations of the United Kingdom on the right side of the figure. Source data are provided as a Source Data file.

In the UK sample, the combined vaccine hesitant and resistant group differed most clearly from the vaccine acceptance group on the following psychological variables: lower levels of trust in health care professionals ($d = 0.39$), scientists ($d = 0.38$), and the state ($d = 0.16$); higher levels of paranoia ($d = 0.27$) and religious beliefs ($d = 0.21$); lower levels of altruism ($d$'s ranged from 0.17 to 0.22); higher levels of social dominance ($d = 0.21$); lower levels of the personality traits agreeableness ($d = 0.22$) and conscientiousness ($d = 0.17$), and higher levels of neuroticism ($d = 0.11$); higher levels of internal locus of control ($d = 0.16$) and belief in chance ($d = 0.17$), and lower levels of beliefs about the role of powerful others ($d = 0.19$); lower cognitive reflection ($d = 0.14$); and more negative attitudes towards migrants ($d = 0.11$).

When comparing the three groups in the UK data, the vaccine resistant group differed from the vaccine hesitant group in terms of higher levels of conspiracy beliefs ($\eta^2 = 0.01$), and lower levels of trust in scientists ($\eta^2 = 0.03$) and health care professionals ($\eta^2 = 0.04$).

**Objective 4. consumption of, and trust in, information regarding COVID-19.** Figures 2, 3 show the levels of consumption of, and trust in, sources of information for each of the vaccine response groups in the Irish sample. Compared to the vaccine accepting respondents, the vaccine resistant respondents consumed significantly ($p < 0.05$) less information about COVID-19 from newspapers, television, radio, and government agencies, and significantly more information from social media. In relation to trust of the different information sources, compared to the vaccine accepting respondents, the vaccine resistant respondents reported significantly ($p < 0.05$) lower levels of trust in information that was disseminated via newspapers, television broadcasts, radio broadcasts, their doctor, other health care professionals, and government agencies.

There were no significant differences in levels of consumption and trust between the vaccine accepting and vaccine hesitant groups in the Irish sample. Compared to vaccine hesitant responders, vaccine resistant individuals consumed significantly less information about the pandemic from television and radio, and had significantly less trust in information disseminated from newspapers, television broadcasts, radio broadcasts, their doctor, other health care professionals, and government agencies.

Figures 4, 5 show the levels of consumption of, and trust in, sources of information for each of the vaccine response groups in the UK sample. The vaccine resistant group consumed significantly ($p < 0.05$) less information about COVID-19 from newspapers and television broadcasts compared to the vaccine accepting group. In relation to trust in the available information, compared to the vaccine accepting respondents, vaccine resistant respondents reported significantly ($p < 0.05$) lower levels of trust in information that was disseminated via newspapers, television broadcasts, radio broadcasts, their doctors, other health care professionals, and government agencies.

There were no significant differences between the vaccine accepting and vaccine hesitant groups regarding levels of consumption or trust in information. Likewise, there were no significant differences in information consumption between the vaccine hesitant and resistant groups, but the vaccine resistant group did have significantly less trust in information sourced from newspapers, radio broadcasts, their doctor, and other health care professionals.

**Discussion**

Similar rates of vaccine hesitance (26% and 25%) and resistance (9% and 6%) were evident in the Irish and UK samples, with only 65% of the Irish population and 69% of the UK population fully willing to accept a COVID-19 vaccine. These findings align with other estimates across seven European nations where 26% of

**Table 2 Sociodemographic, political, and health indicators associations with vaccine hesitancy and resistance in the Irish sample (N = 1041).**

| | Would you accept a COVID-19 vaccine for yourself? | | | | | | | | |
| | (Reference = vaccine acceptance) | | | | | | (Reference = vaccine hesitant) | | |
| | Vaccine hesitant (maybe) | | | Vaccine resistant (no) | | | Vaccine resistant (no) | | |
| | AOR | 95% CIs | | AOR | 95% CIs | | AOR | 95% CIs | |
|---|---|---|---|---|---|---|---|---|---|
| Sex (female) | **1.62** | **1.18** | **2.22** | 1.24 | 0.77 | 2.00 | 0.77 | 0.46 | 1.29 |
| Age 18–24 | 1.74 | 0.84 | 3.60 | 2.49 | 0.78 | 7.88 | 1.43 | 0.41 | 4.96 |
| 25–34 | 1.42 | 0.74 | 2.72 | 2.79 | 0.99 | 7.82 | 1.96 | 0.64 | 6.02 |
| 35–44 | **2.00** | **1.06** | **3.75** | **3.33** | **1.17** | **9.47** | 1.67 | 0.54 | 5.15 |
| 45–54 | 1.35 | 0.71 | 2.57 | 2.49 | 0.87 | 7.16 | 1.84 | 0.59 | 5.79 |
| 55–64 | 1.47 | 0.82 | 2.62 | 1.01 | 0.35 | 2.87 | 0.69 | 0.23 | 2.11 |
| Birthplace (Ireland) | 0.82 | 0.47 | 1.43 | 1.44 | 0.57 | 3.63 | 1.76 | 0.66 | 4.67 |
| Ethnicity (non-Irish) | 1.05 | 0.59 | 1.89 | **2.89** | **1.17** | **7.09** | **2.76** | **1.05** | **7.19** |
| Resident in city | 1.01 | 0.66 | 1.57 | **1.90** | **1.02** | **3.54** | 1.88 | 0.96 | 3.67 |
| Resident in suburb | 0.65 | 0.41 | 1.04 | 0.57 | 0.26 | 1.26 | 0.87 | 0.37 | 2.05 |
| Resident in town | 0.94 | 0.63 | 1.38 | 0.72 | 0.38 | 1.35 | 0.77 | 0.39 | 1.51 |
| No education | 1.14 | 0.81 | 1.61 | 1.25 | 0.75 | 2.10 | 1.10 | 0.63 | 1.92 |
| Unemployed | 0.97 | 0.65 | 1.44 | 0.79 | 0.45 | 1.39 | 0.82 | 0.44 | 1.51 |
| Income €0 – €1999 | 1.05 | 0.63 | 1.75 | **5.73** | **2.19** | **14.96** | **5.44** | **1.98** | **14.93** |
| €20,000 – €29,999 | 1.23 | 0.77 | 1.97 | **3.46** | **1.33** | **9.00** | **2.82** | **1.04** | **7.66** |
| €30,000 – €39,999 | 0.95 | 0.58 | 1.54 | **3.16** | **1.23** | **8.13** | **3.34** | **1.23** | **9.04** |
| €40,000 – €49,999 | 0.92 | 0.53 | 1.58 | **4.79** | **1.82** | **12.64** | **5.24** | **1.86** | **14.73** |
| Only adult in household | 0.71 | 0.47 | 1.05 | 0.61 | 0.34 | 1.08 | 0.86 | 0.46 | 1.60 |
| No children in household | 1.13 | 0.80 | 1.59 | 1.22 | 0.73 | 2.02 | 1.08 | 0.62 | 1.87 |
| Voted Fine Gael | 1.31 | 0.59 | 2.90 | 2.31 | 0.71 | 7.50 | 1.77 | 0.49 | 6.39 |
| Voted Fine Fail | 1.71 | 0.75 | 3.90 | 2.89 | 0.85 | 9.84 | 1.69 | 0.45 | 6.40 |
| Voted Sinn Féin | 1.91 | 0.92 | 3.97 | **3.22** | **1.14** | **9.08** | 1.69 | 0.54 | 5.23 |
| Voted other | 1.23 | 0.55 | 2.75 | 1.64 | 0.51 | 5.23 | 1.33 | 0.37 | 4.74 |
| Voted independent | 2.16 | 0.91 | 5.14 | **4.15** | **1.19** | **14.49** | 1.93 | 0.49 | 7.44 |
| Religiosity (no) | 0.82 | 0.56 | 1.20 | 1.14 | 0.67 | 1.93 | 1.39 | 0.77 | 2.49 |
| Voter (yes) | 0.53 | 0.27 | 1.04 | 0.48 | 0.19 | 1.21 | 0.90 | 0.33 | 2.49 |
| Mental health history (yes) | **0.63** | **0.45** | **0.88** | 0.77 | 0.46 | 1.28 | 1.22 | 0.70 | 2.14 |
| Underlying health condition (present) | 0.97 | 0.61 | 1.54 | **2.59** | **1.38** | **4.85** | **2.68** | **1.33** | **5.38** |
| Underlying health condition – relative | 1.11 | 0.79 | 1.57 | 1.72 | 0.99 | 2.99 | 1.55 | 0.86 | 2.80 |
| Pregnant | 0.78 | 0.36 | 1.68 | 0.92 | 0.33 | 2.62 | 1.19 | 0.39 | 3.66 |
| C-19 infection | 0.61 | 0.30 | 1.25 | 1.92 | 0.40 | 9.18 | 3.16 | 0.65 | 15.42 |
| C-19 infection – relative | 1.03 | 0.56 | 1.89 | 1.64 | 0.54 | 5.02 | 1.59 | 0.49 | 5.16 |

Multinomial logistic regression analyses were performed to identify the key sociodemographic, political, and health-related indicators associated with vaccine hesitancy and resistance. All predictors are adjusted for all other covariates in the model. Note: *AOR* adjusted odds ratios, *95% CIs* 95% confidence intervals for the adjusted odds ratios; statistically significant associations ($p < .05$) are highlighted in bold.

adults indicated hesitance or resistance to a COVID-19 vaccine[7] and in the United States where 33% of the population indicated hesitance or resistance[34]. Rates of resistance to a COVID-19 vaccine also parallel those found for other types of vaccines. For example, in the United States 9% regarded the MMR vaccine as unsafe in a survey of over 1000 adults[35], while 7% of respondents across the world said they "strongly disagree" or "somewhat disagree" with the statement 'Vaccines are safe'[36]. Thus, upwards of approximately 10% of study populations appear to be opposed to vaccinations in whatever form they take. Importantly, however, the findings from the current study and those from around Europe and the United States may not be consistent with or reflective of vaccine acceptance, hesitancy, or resistance in non-Western countries or regions.

**The sociodemographic profile of COVID-19 vaccine hesitant and resistant people.** Across the Irish and UK samples, similarities and differences emerged regarding those in the population who were more likely to be hesitant about, or resistant to, a vaccine for COVID-19. Three demographic factors were significantly associated with vaccine hesitance or resistance in both countries: sex, age, and income level. Compared to respondents accepting of a COVID-19 vaccine, women were more likely to be vaccine hesitant, a finding consistent with a number of studies identifying sex and gender-related differences in vaccine uptake and acceptance[37,38]. Younger age was also related to vaccine hesitance and resistance. However, whereas in the UK all age groups under the age of 65 were more likely to be hesitant or resistant than accepting, only those aged between 35–44 were more likely to be hesitant or resistant in Ireland. Consistent with previous research[39], vaccine resistance was associated with lower income in the UK and Ireland with all earning categories below the highest income bracket associated with COVID-19 vaccine resistance.

Similarity in sociodemographic predictors of COVID-19 vaccine hesitance and resistance across Ireland and the UK may not be considered unusual given their geographical proximity. In Ireland, vaccine resistance was also associated with non-Irish born status, city dwelling, having voted for an anti-establishment or independent candidate in the most recent general election, and

**Table 3 Sociodemographic, political, and health indicators associations with vaccine hesitancy and resistance in the UK sample (N = 2025).**

| | Would you accept a COVID-19 vaccine for yourself? | | | | | | | | |
| | (Reference = vaccine acceptance) | | | | | | (Reference = vaccine hesitant) | | |
| | Vaccine hesitant (maybe) | | | Vaccine resistant (no) | | | Vaccine resistant (no) | | |
| | AOR | 95% CIs | | AOR | 95% CIs | | AOR | 95% CIs | |
|---|---|---|---|---|---|---|---|---|---|
| Sex (female) | **1.43** | **1.14** | **1.80** | 1.05 | 0.69 | 1.60 | 0.73 | 0.47 | 1.15 |
| Age 18–24 | **1.90** | **1.11** | **3.26** | **13.90** | **3.82** | **50.53** | **7.30** | **1.89** | **28.18** |
| 25–34 | **2.33** | **1.46** | **3.72** | **10.11** | **2.89** | **35.34** | **4.34** | **1.18** | **15.91** |
| 35–44 | **2.53** | **1.59** | **4.03** | **11.83** | **3.36** | **41.60** | **4.67** | **1.27** | **17.25** |
| 45–54 | **2.27** | **1.47** | **3.52** | **4.91** | **1.37** | **17.65** | 2.16 | 0.58 | 8.12 |
| 55–64 | **2.48** | **1.60** | **3.85** | **4.36** | **1.19** | **16.00** | 1.76 | 0.46 | 6.74 |
| White British/Irish | 0.83 | 0.35 | 1.93 | 0.94 | 0.26 | 3.39 | 1.14 | 0.28 | 4.62 |
| White other | 0.87 | 0.32 | 2.38 | 0.84 | 0.19 | 3.79 | 0.96 | 0.19 | 4.99 |
| African/Afro-Caribbean | 0.50 | 0.16 | 1.56 | 0.62 | 0.12 | 3.17 | 1.24 | 0.21 | 7.29 |
| Chinese/Asian | 0.66 | 0.20 | 2.12 | 2.16 | 0.20 | 23.83 | 3.30 | 0.27 | 40.33 |
| Indian/Pakistani/Bangladeshi | 0.43 | 0.16 | 1.16 | 1.04 | 0.21 | 5.12 | 2.41 | 0.44 | 13.22 |
| Resident in city | 1.09 | 0.76 | 1.57 | 2.07 | 0.97 | 4.45 | 1.90 | 0.86 | 4.21 |
| Resident in suburb | 1.04 | 0.74 | 1.46 | **2.13** | **1.01** | **4.49** | 2.05 | 0.94 | 4.45 |
| Resident in town | 0.88 | 0.62 | 1.23 | 1.35 | 0.63 | 2.89 | 1.55 | 0.70 | 3.41 |
| No education | 0.53 | 0.29 | 0.98 | 0.63 | 0.20 | 1.98 | 1.19 | 0.37 | 3.80 |
| Unemployed | 0.72 | 0.51 | 1.00 | 1.06 | 0.56 | 1.98 | 1.48 | 0.77 | 2.84 |
| Income 0–£15,490 | 1.20 | 0.81 | 1.80 | **2.48** | **1.11** | **5.54** | 2.07 | 0.89 | 4.80 |
| £15,491–£25,340 | 1.31 | 0.90 | 1.90 | **2.68** | **1.28** | **5.63** | 2.05 | 0.94 | 4.49 |
| £25,341–£38,740 | 1.31 | 0.91 | 1.88 | **2.31** | **1.10** | **4.84** | 1.76 | 0.81 | 3.84 |
| £38,741–£57,930 | 1.10 | 0.77 | 1.56 | 1.17 | 0.52 | 2.62 | 1.06 | 0.46 | 2.47 |
| Only adult in household | 0.81 | 0.61 | 1.09 | 1.05 | 0.62 | 1.77 | 1.29 | 0.74 | 2.23 |
| No children in the household | 1.15 | 0.88 | 1.49 | 0.97 | 0.61 | 1.53 | 0.84 | 0.52 | 1.37 |
| Voter (no) | 0.62 | 0.42 | 0.90 | 0.73 | 0.39 | 1.38 | 1.19 | 0.61 | 2.31 |
| Brexit (leave) | 1.08 | 0.85 | 1.38 | 1.57 | 0.99 | 2.47 | 1.45 | 0.89 | 2.34 |
| Religiosity (no) | 0.92 | 0.73 | 1.16 | 0.78 | 0.50 | 1.21 | 0.85 | 0.54 | 1.36 |
| Mental health history (yes) | 0.90 | 0.71 | 1.15 | 0.83 | 0.54 | 1.27 | 0.92 | 0.59 | 1.44 |
| Underlying health condition (present) | 1.32 | 0.94 | 1.85 | 0.91 | 0.50 | 1.65 | 0.69 | 0.36 | 1.30 |
| Underlying health condition – relative | 1.07 | 0.81 | 1.40 | 1.30 | 0.77 | 2.20 | 1.22 | 0.70 | 2.11 |
| Pregnant | 1.26 | 0.69 | 2.27 | **2.36** | **1.03** | **5.40** | 1.88 | 0.78 | 4.57 |
| C-19 infection | 1.17 | 0.67 | 2.04 | 0.69 | 0.30 | 1.60 | 0.59 | 0.24 | 1.47 |
| C-19 infection – relative | 0.81 | 0.51 | 1.29 | 1.68 | 0.62 | 4.54 | 2.07 | 0.74 | 5.78 |

Multinomial logistic regression analyses were performed to identify the key sociodemographic, political, and health-related indicators associated with vaccine hesitancy and resistance. All predictors are adjusted for all other covariates in the model. Note: AOR adjusted odds ratios, 95% CIs 95% confidence intervals for the adjusted odds ratios; statistically significant associations (p < .05) are highlighted in bold.

having an underlying chronic health problem; while in the UK, vaccine resistance was associated with suburban dwelling and being pregnant. Urban/suburban dwelling may reflect broader socioeconomic issues known to underpin vaccine hesitancy[40–43], and is a worrying finding given the greater potential for community transmission within more densely populated areas. Vaccine uptake among minority groups is often lower than that among the general population[44–46], and the reasons for this disparity may include limited access to primary care, failure of clinical staff to communicate the importance of vaccination during health care visits, and/or misconceptions about the costs, adverse effects, risks, and benefits of vaccination[47–49]. Greater resistance to vaccination seen in populations with existing chronic health problems may be explained by the presence of individuals for whom vaccines are medically contraindicated or by a fear of iatrogenic effects of a vaccine among these individuals[50–52]. Pregnancy has also been found to be associated with increased resistance to vaccines for other communicable diseases, such as influenza and pertussis[53–55].

Taken together, our findings show that although there are some similarities regarding who in the Irish and UK populations are most likely to be hesitant about, or resistant to, a potential COVID-19 vaccine, many of the determining factors are likely to be context-dependent. Therefore, national public health authorities can use these findings in two ways. First, based on the common risk factors for vaccine hesitance/resistance across the samples, public health campaigns could be targeted at groups more likely to be vaccine hesitant or resistant, including women, younger adults, and those of lower socioeconomic status. Second, based on the unique risk factors for vaccine hesitance/resistance across the samples, public health authorities in different nations should seek to replicate our work with a view to identifying the characteristics of vaccine hesitant or resistant sub-groups within their own contexts, and direct public health messaging to specifically target these groups. A multi-disciplinary approach engaging social and behavioural change communication experts, social marketers, medical anthropologists, psychologists, and health care practitioners is likely to be required.

**The psychological profile of COVID-19 vaccine hesitant and resistant people**. Interestingly, while vaccine hesitant and

**Table 4 Psychological indicators of vaccine acceptance/hesitancy/resistance in the Irish sample.**

| | Vaccine acceptance [a] | | | Vaccine hesitance [b] | | | Vaccine resistance [c] | | | | Hesitance & Resistance [d] | | | |
|---|---|---|---|---|---|---|---|---|---|---|---|---|---|---|
| | n = 665 (65%) | | | n = 262 (26%) | | | n = 9CCC | | | | n = 359 (35%) | | | |
| | Mean | SD | SE | Mean | SD | SE | Mean | SD | SE | η2 | Mean | SD | SE | d |
| *Personality* | | | | | | | | | | | | | | |
| Extraversion | 6.05 | 1.90 | 0.07 | 6.20 | 1.84 | 0.11 | 6.25 | 2.04 | 0.21 | 0.002 | 6.21 | 1.89 | 0.10 | 0.08 |
| Agreeableness | 7.08 *d* | 1.57 | 0.06 | 6.84 | 1.64 | 0.10 | 6.87 | 1.61 | 0.16 | 0.005 | 6.84 *a* | 1.63 | 0.09 | **0.15** |
| Conscientiousness | 7.48 | 1.72 | 0.07 | 7.52 | 1.72 | 0.11 | 7.57 | 1.66 | 0.17 | 0.000 | 7.53 | 1.70 | 0.09 | 0.03 |
| Neuroticism | 5.55 | 2.11 | 0.08 | 5.52 | 1.94 | 0.12 | 5.66 | 2.15 | 0.22 | 0.000 | 5.56 | 1.99 | 0.11 | 0.01 |
| Openness | 6.62 | 1.65 | 0.06 | 6.59 | 1.62 | 0.10 | 6.90 | 1.61 | 0.16 | 0.002 | 6.67 | 1.62 | 0.09 | 0.03 |
| *LOC* | | | | | | | | | | | | | | |
| Chance | 11.19 | 3.76 | 0.15 | 11.15 | 3.10 | 0.19 | 10.77 | 4.09 | 0.42 | 0.001 | 11.04 | 3.39 | 0.18 | 0.04 |
| Powerful others | 9.77 | 4.09 | 0.16 | 9.60 | 3.69 | 0.23 | 10.19 | 4.14 | 0.42 | 0.001 | 9.76 | 3.82 | 0.20 | 0.003 |
| Internal | 8.77 *d* | 3.22 | 0.12 | 9.20 | 2.89 | 0.18 | 9.24 | 3.42 | 0.35 | 0.004 | 9.21 *a* | 3.03 | 0.16 | **0.14** |
| *CRT* | | | | | | | | | | | | | | |
| Tests 1–3 | **0.88** *d* | 1.09 | 0.04 | 0.76 | 0.97 | 0.06 | 0.70 | 0.99 | 0.10 | 0.004 | **0.74** *a* | 0.97 | 0.05 | **0.25** |
| *Altruism* | | | | | | | | | | | | | | |
| Identify with others | 11.07 *bd* | 2.43 | 0.09 | 10.52 *a* | 2.38 | 0.15 | 10.44 | 2.59 | 0.26 | **0.012** | 10.50 *a* | 2.43 | 0.13 | **0.24** |
| Care about others | 11.59 *bd* | 2.53 | 0.10 | 10.92 *a* | 2.69 | 0.17 | 11.22 | 2.73 | 0.28 | **0.013** | 11.00 *a* | 2.70 | 0.14 | **0.23** |
| Help others | 11.40 *bd* | 2.50 | 0.10 | 10.93 *a* | 2.62 | 0.16 | 11.03 | 2.84 | 0.29 | **0.007** | 10.96 *a* | 2.68 | 0.14 | **0.17** |
| *Beliefs* | | | | | | | | | | | | | | |
| Religious | 25.23 *bd* | 6.43 | 0.25 | 24.08 *a* | 5.70 | 0.36 | 23.78 | 6.39 | 0.66 | **0.009** | 23.99 *a* | 5.89 | 0.32 | **0.20** |
| Conspiracy | 35.59 *cd* | 9.12 | 0.35 | 36.60 *c* | 9.24 | 0.57 | 40.20 *ab* | 9.50 | 0.96 | **0.021** | 37.57 *a* | 9.44 | 0.50 | **0.21** |
| Paranoia | 12.05 | 5.02 | 0.20 | 12.32 | 4.37 | 0.27 | 13.18 | 5.04 | 0.51 | 0.005 | 12.55 | 4.57 | 0.24 | 0.10 |
| *Trust* | | | | | | | | | | | | | | |
| State | 14.33 *bcd* | 4.44 | 0.17 | 13.29 *a* | 4.07 | 0.25 | 12.16 *a* | 4.70 | 0.48 | **0.025** | 12.98 *a* | 4.27 | 0.23 | **0.31** |
| Scientists | 3.76 *bcd* | 0.91 | 0.04 | 3.30 *a* | 1.02 | 0.06 | 3.21 *a* | 1.15 | 0.12 | **0.056** | 3.27 *a* | 1.05 | 0.06 | **0.51** |
| Health care profs | 3.95 *bcd* | 0.93 | 0.04 | 3.57 *a* | 0.99 | 0.06 | 3.36 *a* | 1.20 | 0.12 | **0.047** | 3.51 *a* | 1.05 | 0.06 | **0.45** |
| *Socio-political views* | | | | | | | | | | | | | | |
| Authoritarianism | 17.80 *d* | 3.99 | 0.16 | 18.42 | 3.45 | 0.21 | 18.09 | 4.06 | 0.41 | **0.005** | 18.33 *a* | 3.62 | 0.19 | **0.14** |
| Social dominance | 17.94 *bd* | 5.34 | 0.21 | 19.15 *a* | 4.83 | 0.30 | 18.87 | 5.25 | 0.16 | **0.011** | 19.08 *a* | 5.05 | 0.27 | **0.22** |
| Migrant views 1 | 6.52 *bd* | 2.37 | 0.09 | 5.86 *a* | 2.19 | 0.14 | 5.92 | 2.44 | 0.25 | **0.017** | 5.88 *a* | 2.26 | 0.12 | **0.27** |
| Migrant views 2 | 6.57 *bcd* | 2.49 | 0.10 | 5.85 *a* | 2.24 | 0.14 | 5.90 *a* | 2.50 | 0.25 | **0.019** | 5.86 *a* | 2.31 | 0.12 | **0.29** |

Note: ***abcd*** = mean difference between denoted categories is significant at the <0.001 level. *abcd* = mean difference between denoted categories is significant at 0.05 level. Statistically significant comparisons is significant at 0.05 to 0.001 level. Statistically significant comparisons in bold. Column *d* is a two-tailed independent samples t-test.

**Table 5 Psychological indicators of vaccine acceptance/hesitancy/resistance in the UK sample.**

| | Vaccine acceptance [a] | | | Vaccine hesitance [b] | | | Vaccine resistance [c] | | | | Hesitance & resistance [d] | | | |
|---|---|---|---|---|---|---|---|---|---|---|---|---|---|---|
| | n = 1383 (69%) | | | n = 501 (25%) | | | n = 124 (6%) | | | | n = 625 (31%) | | | |
| | Mean | SD | SE | Mean | SD | SE | Mean | SD | SE | $\eta2$ | Mean | SD | SE | d |
| **Personality** | | | | | | | | | | | | | | |
| Extraversion | 5.55 | 1.80 | 0.05 | 5.39 | 1.61 | 0.07 | 5.47 | 1.77 | 0.16 | 0.002 | 5.41[a] | 1.64 | 0.07 | 0.08 |
| Agreeableness | 6.85[bcd] | 1.61 | 0.04 | 6.58[a] | 1.53 | 0.07 | 6.23[a] | 1.76 | 0.16 | 0.012 | 6.50[a] | 1.58 | 0.06 | 0.22 |
| Conscientiousness | 7.55[bcd] | 1.73 | 0.05 | 7.31[a] | 1.72 | 0.08 | 7.03[a] | 1.87 | 0.17 | 0.007 | 7.25[a] | 1.75 | 0.07 | 0.17 |
| Neuroticism | 5.63[d] | 2.15 | 0.06 | 5.83 | 2.01 | 0.09 | 5.98 | 2.08 | 0.19 | 0.003 | 5.86[a] | 2.03 | 0.08 | 0.11 |
| Openness | 6.52 | 1.69 | 0.05 | 6.48 | 1.64 | 0.07 | 6.52 | 1.44 | 0.13 | 0.000 | 6.45 | 1.60 | 0.06 | 0.04 |
| **LOC** | | | | | | | | | | | | | | |
| Chance | 11.25[bcd] | 3.81 | 0.10 | 11.83[a] | 3.35 | 0.15 | 12.12[a] | 3.89 | 0.35 | 0.007 | 11.89[a] | 3.46 | 0.14 | 0.17 |
| Powerful others | 14.14[bcd] | 4.26 | 0.11 | 13.52[a] | 4.02 | 0.18 | 12.83[a] | 4.49 | 0.40 | 0.008 | 13.38 | 4.12 | 0.16 | 0.18 |
| Internal | 9.11[bcd] | 3.16 | 0.08 | 9.77[a] | 3.08 | 0.14 | 10.06[a] | 3.84 | 0.35 | 0.011 | 9.83 | 3.25 | 0.13 | 0.22 |
| **CRT** | | | | | | | | | | | | | | |
| Tests 1–3 | .96[d] | 1.09 | 0.03 | 0.84 | 1.02 | 0.05 | 0.73 | 0.99 | 0.09 | 0.004 | 0.81[a] | 1.01 | 0.04 | 0.14 |
| **Altruism** | | | | | | | | | | | | | | |
| Identify with others | 10.06[bcd] | 2.55 | 0.07 | 9.68[a] | 2.50 | 0.11 | 9.43[a] | 3.03 | 0.27 | 0.006 | 9.63[a] | 2.62 | 0.10 | 0.17 |
| Care about others | 10.54[bcd] | 2.73 | 0.07 | 10.14[a] | 2.70 | 0.12 | 9.63[a] | 3.17 | 0.28 | 0.009 | 10.04[a] | 2.81 | 0.11 | 0.18 |
| Help others | 10.30[bcd] | 2.77 | 0.07 | 9.80[a] | 2.62 | 0.12 | 9.27[a] | 3.11 | 0.28 | 0.012 | 9.69[a] | 2.73 | 0.11 | 0.22 |
| **Beliefs** | | | | | | | | | | | | | | |
| Religious | 27.84[bcd] | 6.37 | 0.17 | 26.75[a] | 5.65 | 0.25 | 25.73[a] | 5.83 | 0.53 | 0.011 | 26.55[a] | 5.70 | 0.23 | 0.21 |
| Conspiracy | 35.07[c] | 9.07 | 0.24 | 34.47[c] | 9.00 | 0.40 | 38.73[ab] | 10.02 | 0.90 | 0.011 | 35.31 | 9.36 | 0.37 | 0.03 |
| Paranoia | 12.04[bc] | 5.02 | 0.14 | 13.13[a] | 4.73 | 0.21 | 14.27[a] | 4.81 | 0.43 | 0.018 | 13.36[a] | 4.77 | 0.19 | 0.27 |
| **Trust** | | | | | | | | | | | | | | |
| State | 13.86[cd] | 4.12 | 0.11 | 13.36 | 3.91 | 0.18 | 12.68[a] | 4.92 | 0.44 | 0.006 | 13.22[a] | 4.13 | 0.17 | 0.16 |
| Scientists | 3.77[bcd] | 0.96 | 0.03 | 3.45[ac] | 0.98 | 0.04 | 3.20[ab] | 1.08 | 0.10 | 0.033 | 3.40[a] | 1.00 | 0.04 | 0.38 |
| Health care profs | 4.01[bcd] | 0.95 | 0.03 | 3.71[ac] | 0.97 | 0.04 | 3.32[ab] | 1.11 | 0.10 | 0.039 | 3.63[a] | 1.01 | 0.04 | 0.39 |
| **Socio-political views** | | | | | | | | | | | | | | |
| Authoritarianism | 29.62 | 6.76 | 0.18 | 29.46 | 6.18 | 0.28 | 29.60 | 6.54 | 0.59 | 0.000 | 29.48 | 6.25 | 0.25 | 0.02 |
| Social dominance | 23.45[bd] | 8.83 | 0.24 | 25.03[a] | 8.65 | 0.39 | 26.50[a] | 8.99 | 0.81 | 0.011 | 25.32[a] | 8.73 | 0.35 | 0.21 |
| Migrant views 1 | 6.34 | 2.31 | 0.06 | 6.21 | 2.25 | 0.10 | 6.16 | 2.41 | 0.22 | 0.001 | 6.20 | 2.28 | 0.09 | 0.06 |
| Migrant views 2 | 6.16[d] | 2.53 | 0.07 | 5.92 | 2.45 | 0.11 | 5.71 | 2.54 | 0.23 | 0.003 | 5.88[a] | 2.47 | 0.10 | 0.11 |

Note: **abcd** = mean difference between denoted categories is significant at the 0.001 level. abcd = mean difference between denoted categories is significant at the 0.05 to 0.001 level. Statistically significant comparisons in bold. Column d is a two-tailed independent samples t-test.

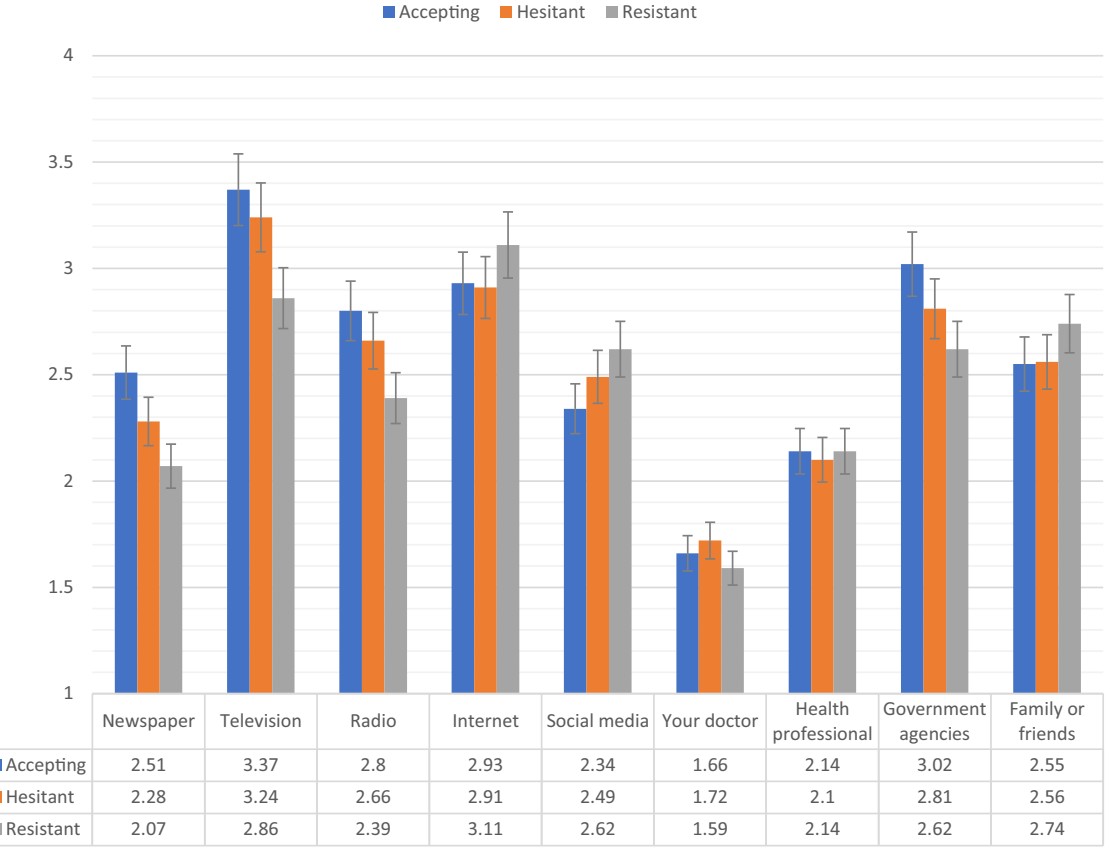

**Fig. 2 Sources of COVID-19 information for vaccine accepting, hesitant, and resistant groups in the Irish sample.** Data presented show the degree to which COVID-19 vaccine accepting (blue), hesitant (orange), and resistant (grey) respondents from the Irish sample (N = 1041) source information about COVID-19 from nine separate sources. Scaling on y-axis denotes 1–4 Likert scaling of 'Sources of COVID-19 Information' measure (1 = none, 2 = a little, 3 = some, 4 = a lot). Error bars present the 95% confidence intervals of these proportions. Source data are provided as a Source Data file.

resistant individuals in Ireland and the UK varied in relation to their social, economic, cultural, political, and geographical characteristics, both populations shared similar psychological profiles. Specifically, COVID-19 vaccine hesitant or resistant persons were distinguished from their vaccine accepting counterparts by being more self-interested, more distrusting of experts and authority figures (i.e. scientists, health care professionals, the state), more likely to hold strong religious beliefs (possibly because these kinds of beliefs are associated with distrust of the scientific worldview) and also conspiratorial and paranoid beliefs (which reflect lack of trust in the intentions of others). They were also more likely to believe that their lives are primarily under their own control, to have a preference for societies that are hierarchically structured and authoritarian, and to be more intolerant of migrants in society (attitudes that have been previously hypothesised to be consistent with, and understandable in the context of, evolved responses to the threat of pathogens)[56]. They were also more impulsive in their thinking style, and had a personality characterised by being more disagreeable, more emotionally unstable, and less conscientious.

**Reaching those who are hesitant or resistant to a COVID-19 vaccine.** Responsibility for public health messaging primarily lies with governments, scientists, and medical professionals. The high level of distrust that vaccine hesitant and resistant people have for those who represent established authority is likely to provoke psychological resistance to any message emanating from these sources, and to an entrenchment of their existing 'anti-establishment' or 'anti-authority' beliefs. Consequently, anti-vaccine

beliefs may be expressed by some individuals in society as a way to advertise their 'anti-establishment' sentiments. By understanding the psychological dispositions of these individuals, another – potentially more effective – approach could be adopted. For example, recognising their preference for social dominance and authoritarianism, and their distrust of conventional authority figures, vaccine hesitant or resistant persons may be more receptive to authoritative messages regarding COVID-19 vaccine safety and efficacy if they are delivered by individuals within non-traditional positions of authority and expertise. Engagement of religious leaders, for example, has been documented as an important approach to improve vaccine acceptance[16,57]. Key to the preparation of a COVID-19 vaccine is, therefore, the early and frequent engagement of religious and community-leaders[58], and for health authorities to work collaboratively with multiple societal stakeholders to avoid the feeling that they are only acting on behalf of government authorities[59].

Moreover, given their lack of altruism, their internal locus of control, and their anti-migrant views, messages tailored to vaccine hesitant or resistant individuals could emphasise the personal benefits of vaccination against COVID-19, and the benefits to those with whom they most closely identify. Furthermore, given that the results of this study indicate that vaccine hesitant or resistant individuals are typically less agreeable, less conscientious, less emotionally stable, and less analytically capable, public health messaging targeted at these persons should be clear, direct, repeated, and positively orientated.

Aligned to our findings that vaccine resistant individuals were more distrusting of scientific expertise and health and

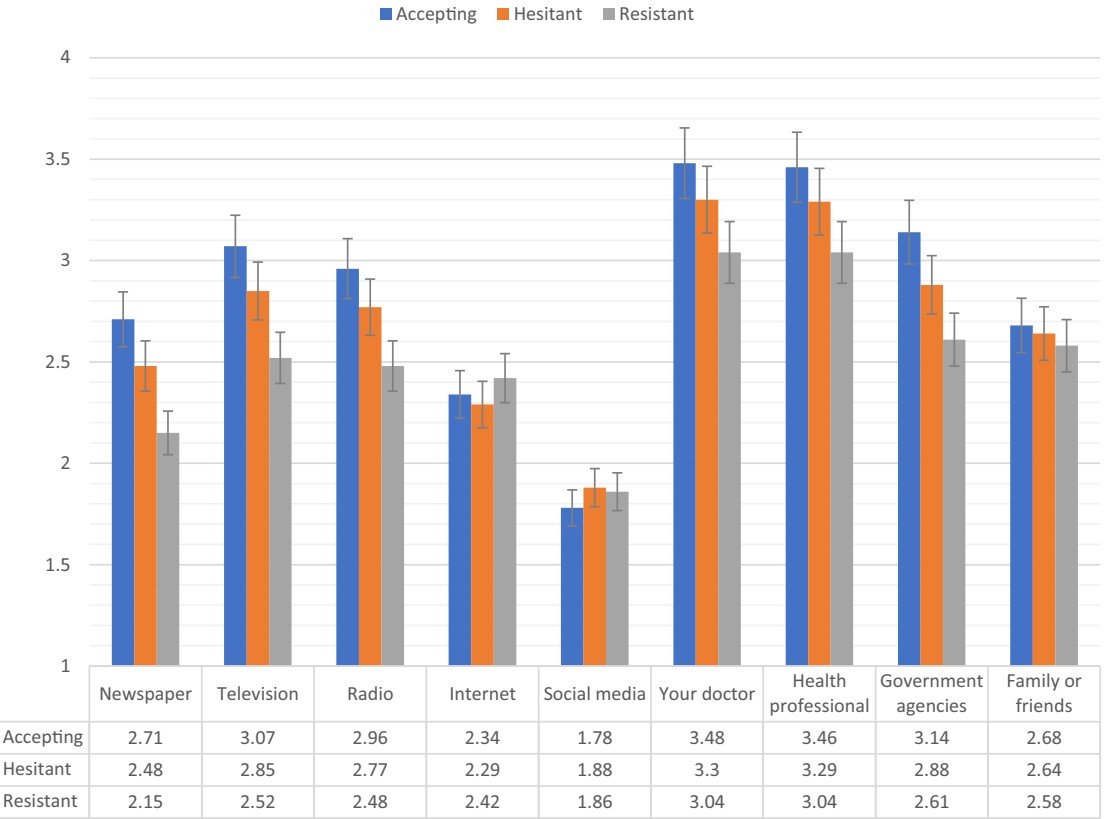

**Fig. 3 Trust in COVID-19 information sources for vaccine accepting, hesitant, and resistant groups in the Irish sample.** Data presented show the degree to which COVID-19 vaccine accepting (blue), hesitant (orange), and resistant (grey) respondents from the Irish sample (N = 1041) trust information about COVID-19 from nine separate sources. Scaling on y-axis denotes 1–4 Likert scaling of 'Sources of COVID-19 Information' measure (1 = none, 2 = a little, 3 = some, 4 = a lot). Error bars present the 95% confidence intervals of these proportions. Source data are provided as a Source Data file.

government authorities, vaccine resistant individuals in both countries were less likely to consume, and trust, information from 'traditional' sources (i.e. newspapers, television, radio, and government agencies), and were somewhat more likely to obtain information from social media channels. These findings are consistent with global trends and other studies reporting social media as an instrumental platform for anti-vaccine messaging[60,61]. This poses further challenges to effective communication with vaccine resistant individuals, and highlights the need for public health officials to disseminate information via multiple media channels to increase the chances of accessing vaccine resistant or hesitant individuals. Knowledge of the sociodemographic and psychological profiles of vaccine hesitant/resistant individuals, combined with knowledge of what information sources they access, and whom they trust most, provides important information for public health officials to effectively design and deliver public health messages so that a sufficient proportion of the population will voluntarily accept a vaccine for COVID-19.

**Study limitations**. These findings should be interpreted in light of several limitations. First, quota sampling was used to recruit both non-probability-based samples via the internet. This opt-in mode of recruitment employed by the survey company who facilitated the data collection (Qualtrics), albeit being a cost-effective method for gaining fast access to a large and diverse sample (and largely the only feasible method of recruitment during the pandemic), inevitably meant that it was not possible to generate a response rate for the baseline survey due to the lack of a known denominator or sampling frame. Whilst more research is required to fully investigate the strengths and weaknesses

associated with internet-based panel surveying[62], it has been suggested that the composition of non-probability internet-based survey panels differs from that of the underlying population[63]. Indeed, the American Association for Public Opinion Research (APPOR) asserts that when non-probability sampling methods are used, there is a higher burden of responsibility on investigators to describe the methods used to draw the sample and collect the data, so that users can make an informed decision about the usefulness of the resulting survey estimates[64]. We support the APPOR's position that it is useful to think of different non-probability sampling approaches as falling on a continuum of expected accuracy of the survey estimates; at one end are uncontrolled convenience samples that produce risky survey estimates by assuming that respondents are a random sample of the population, whereas at the other end, there are surveys that recruit respondents based on criteria related to the survey subject matter and then the survey results are adjusted using variables that are correlated with the key study outcome variables[64]. The design of the current studies ensures that it falls towards the latter end of this continuum.

Second, these data were collected during the first week of the strictest lockdown measures that had ever been imposed in Ireland and the UK, respectively. Thus, rates of vaccine acceptance, hesitance, and resistance will have been affected by these social circumstances. Third, questions were answered with regards to a hypothetical vaccine whose effectiveness, risk of adverse side-effects, and contraindications were unknown. Continued monitoring throughout the pandemic, and throughout the development of the vaccine(s) for COVID-19, will help us to better understand changing levels of hesitance and resistance to vaccination, and our group are engaging in this work.

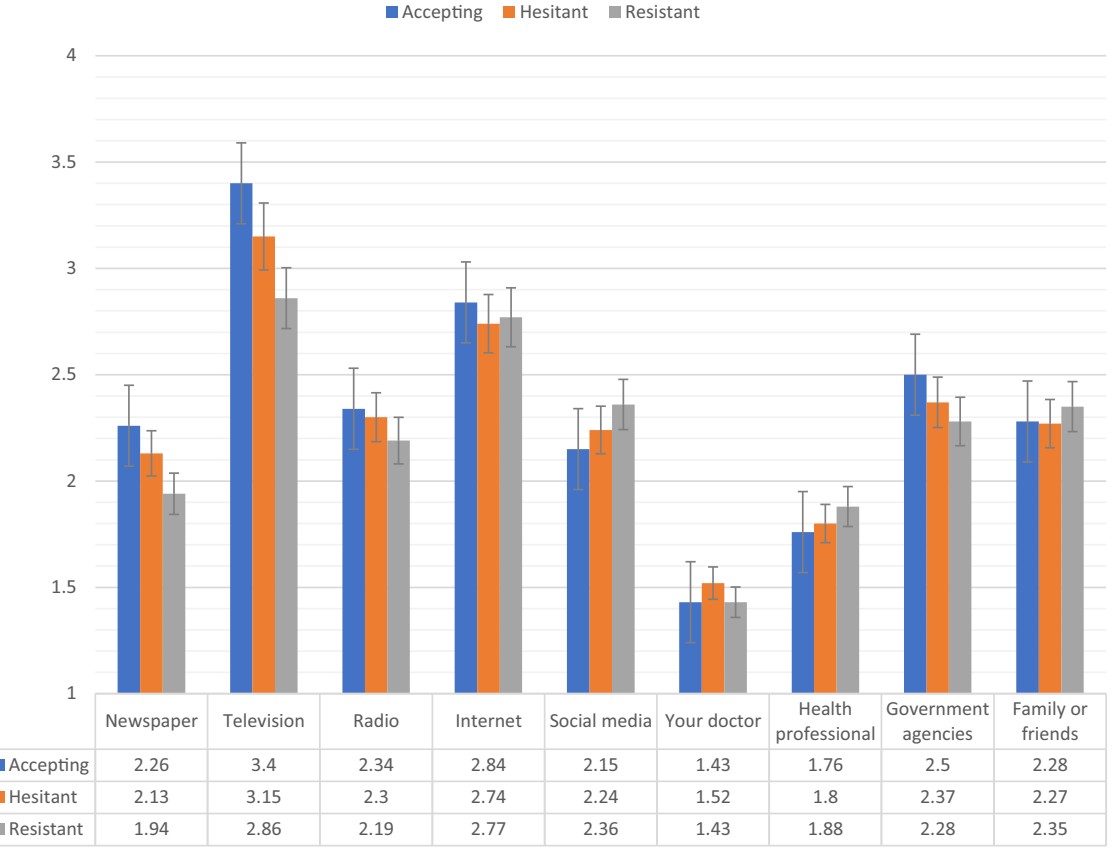

| | Newspaper | Television | Radio | Internet | Social media | Your doctor | Health professional | Government agencies | Family or friends |
|---|---|---|---|---|---|---|---|---|---|
| Accepting | 2.26 | 3.4 | 2.34 | 2.84 | 2.15 | 1.43 | 1.76 | 2.5 | 2.28 |
| Hesitant | 2.13 | 3.15 | 2.3 | 2.74 | 2.24 | 1.52 | 1.8 | 2.37 | 2.27 |
| Resistant | 1.94 | 2.86 | 2.19 | 2.77 | 2.36 | 1.43 | 1.88 | 2.28 | 2.35 |

**Fig. 4 Sources of COVID-19 information for vaccine accepting, hesitant, and resistant groups in the UK sample.** Data presented show the degree to which COVID-19 vaccine accepting (blue), hesitant (orange), and resistant (grey) respondents from the UK sample ($N = 2025$) source information about COVID-19 from nine separate sources. Scaling on y-axis denotes 1–4 Likert scaling of 'Sources of COVID-19 Information' measure (1 = none, 2 = a little, 3 = some, 4 = a lot). Error bars present the 95% confidence intervals of these proportions. Source data are provided as a Source Data file.

Fourth, the current study was also limited to two western, European countries, whose populations had many social, cultural, economic, and political similarities. Relatedly, the extent to which these results will generalise to other nations is unknown, though the similarity of results – especially with respect to the psychological profiles we have identified – in at least two different countries is promising. It is essential that many other (low, middle, and high income) countries obtain estimates of hesitancy/resistance to COVID-19 vaccination in the general population. As is abundantly clear, the spread of the virus does not respect national borders and only a global vaccination programme will lead to success. Nations across the world could potentially prepare for the delivery of a COVID-19 vaccine by identifying psychological characteristics associated with hesitancy and resistance in their populations and honing their public messaging in order to maximise vaccine uptake.

Finally, while the use of nationally representative samples from two countries is a key strength, these samples are representative of general adult populations and do not include members of the public that are institutionalised (e.g. hospital care, prisons, refugee centres) or difficult to reach (e.g. those not online, the homeless, etc.). The inability to survey these members of society also limits the generalisability of our results.

Despite these limitations, our findings provide important evidence regarding the level of hesitance and resistance toward a potential COVID-19 vaccine in two general population samples. The development of a vaccine for COVID-19 represents an enormous ongoing global scientific and political effort; however, our findings suggest that if this global effort is successful and a vaccine is delivered, governments and health workers in many countries are likely to face another battle: how to persuade a sufficient proportion of their populations to accept the vaccine to effectively suppress the virus. We offer these findings in the hope that they highlight the importance of understanding the various social, economic, political, and psychological factors that contribute to COVID-19 vaccine hesitance and resistance, and how they can be used to maximise the positive effect of public health messaging. Convincing members of the public who are hesitant or resistant to a COVID-19 vaccine will require the concerted efforts of multiple stakeholders in society, many of whom are often excluded from mainstream politics and health policy[65]. The engagement and participation of these key and trusted community actors will likely be required to effectively reach and convince a sufficient proportion of individuals in the general population of the necessity of COVID-19 immunisation.

## Methods

**Participants and procedure**. Data from nationally representative samples of the general adult populations of Ireland ($N = 1041$) and the UK ($N = 2025$) were collected by the survey company Qualtrics. These data were collected as part of the COVID-19 Psychological Research Consortium (C19PRC) Study[66] to track the mental health and societal impact of the pandemic across both countries. Quota sampling was used to ensure that the sample characteristics of sex, age, and geographical distribution matched known population parameters for the Irish population, while age, sex and income matched known population parameters for the UK population. The UK data collection took place between March 23rd and 28th, 2020. Data collection began 52 days after the first confirmed case of COVID-19 in the UK, and the same day the UK Prime Minister announced that people were required to stay at home except for very limited purposes. The Irish data collection took place between March 31st and April 5th, 2020. This was 31 days after the first

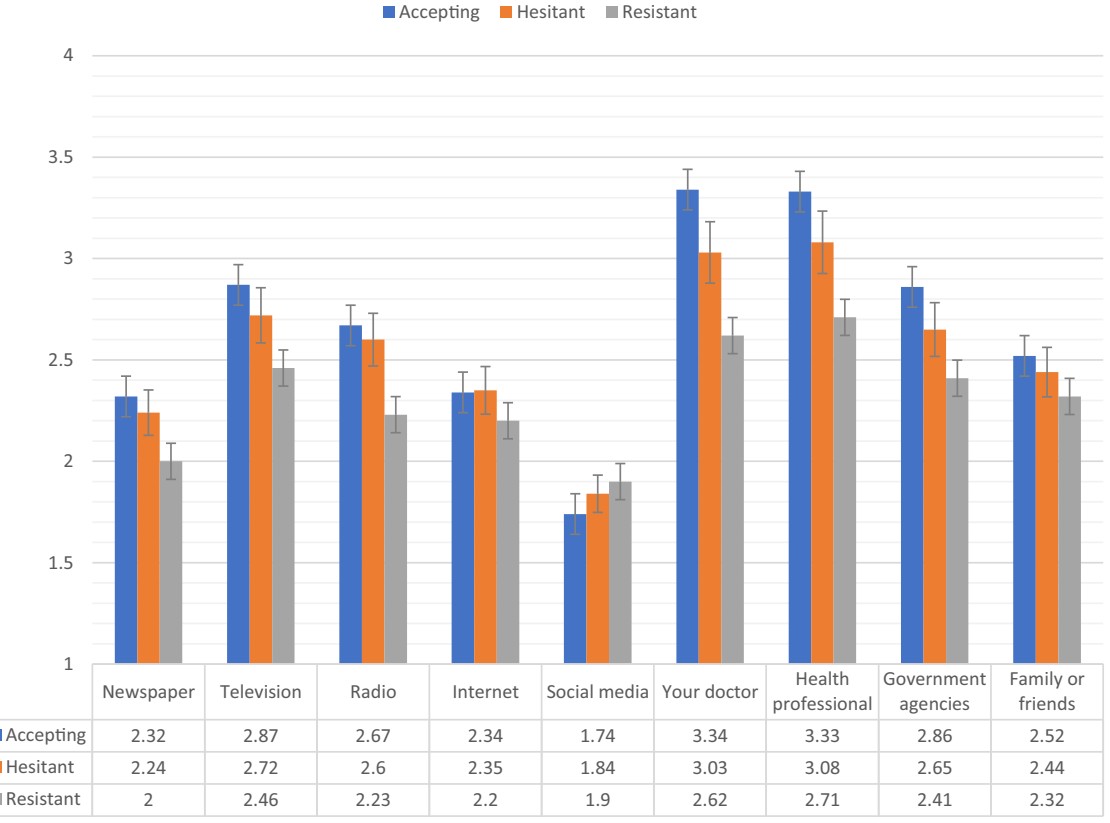

**Fig. 5 Trust in COVID-19 information sources for vaccine accepting, hesitant, and resistant groups in the UK sample.** Data presented show the degree to which COVID-19 vaccine accepting (blue), hesitant (orange), and resistant (grey) respondents from the UK sample ($N = 2025$) trust information about COVID-19 from nine separate sources. Scaling on y-axis denotes 1–4 Likert scaling of 'Sources of COVID-19 Information' measure (1 = none, 2 = a little, 3 = some, 4 = a lot). Error bars present the 95% confidence intervals of these proportions. Source data are provided as a Source Data file.

confirmed case of COVID-19 in Ireland, 19 days after the first physical distancing measures were enacted (i.e. closure of all childcare and educational facilities), and two days after the Taoiseach (Irish Prime Minister) announced that people were not to leave their homes except for very limited purposes. Therefore, these data were collected within the first week of the strictest physical distancing measures being enacted in both countries.

Power analyses were conducted to determine the optimal sample sizes for both countries. As the C19PRC Study was primarily concerned with tracking mental health disorders (depression, generalised anxiety disorder [GAD], and posttraumatic stress disorder [PTSD]) in the general population, sample size calculations were based on existing prevalence estimates for these disorders. In Ireland and the UK, the estimated prevalence of PTSD is 5% and 4%, respectively, and lower than the prevalence estimates of depression and generalised anxiety[67]. To detect a disorder with a prevalence of 4%, with precision of 1%, and 95% confidence level, a sample size of 1476 was required. The survey company used to collect the data could only guarantee a maximum sample size of 1000 participants in Ireland, whereas a larger sample could be obtained in the UK. This is a consequence of the much smaller population of Ireland (4.9 million people) compared to the UK (66.7 million people). Therefore, the target sample size in Ireland was set at 1000 which, holding all other parameters in the sample size calculation equal, resulted in a precision of 1.21%. In the UK, a target sample was set at 2000 to increases the number of 'cases' detected because of the intention to track changes in the mental health problems in the population over time.

Inclusion criteria for both samples were that participants be aged 18 years or older at the time of the survey, resident in the country that the survey was conducted, and be able to complete the survey in English. Participants were contacted by the survey company via email and requested to participate. If consenting, participants completed the survey online (median time of completion = 37.52 and 28.91 min for the Irish and UK surveys, respectively) and were reimbursed by the survey company for their time. Ethical approval for the study was provided by the Research Governance Committee at University of Sheffield (Reference number: 033759) and approved by the School of Psychology Ethics Filter Committee at Ulster University (Reference number: 230320). The sociodemographic characteristics for both samples are reported in Table 1.

**Measures.** Given the distinct socio-political contexts of Ireland and the UK, some variation existed in the measurement of the sociodemographic and political

variables used in this study. All other variables were measured in an identical manner across the two samples.

**COVID-19 vaccination status.** Participants were asked, 'If a new vaccine were to be developed that could prevent COVID-19, would you accept it for yourself?' and were classified as 'vaccine accepting' if they responded 'Yes', 'vaccine hesitant' if they responded 'Maybe', and 'vaccine resistant' if they responded 'No'.

**Sociodemographic, political, and religious indicators.** The sociodemographic variables used in this study were directly informed by the extant evidence base relating to vaccine hesitancy and are outlined in Table 5. In compliance with the International Journal of Epidemiology guidelines on forming age categories we categorised age from mid-decade to mid-decade (e.g. 35–44, 45–54)[68]. Additionally, the Irish and UK samples were asked about their voting behaviours in recent political elections. In the Irish sample, people were asked if they voted in the February 2020 General Election (0 = No, 1 = Yes), and to which political party they gave their first preference vote (Fine Gael, Fianna Fáil, Sinn Féin, Other party, Independents; 0 = No, 1 = Yes). In the UK sample, people were asked if they voted in the most recent general election (0 = No, 1 = Yes), and how they voted in the 2016 European Union membership referendum (0 = Remain, 1 = Leave). In both samples, respondents were asked "What is your religious conviction (how you would classify your religious belief now)?", with response options including: Christian, Muslim, Jewish, Hindu, Buddhist, Sikh, Atheist, Agnostic, Other. A binary variable was generated to represent 'Religion' where 1 = No religion (combining atheist/agnostic?) and 0 = 'Other' (combining all other categories).

**Health-related indicators.** Participants were asked if they have diabetes, lung disease, or heart disease (0 = No, 1 = Yes), if any immediate family members have diabetes, lung disease, or heart disease (0 = No, 1 = Yes), if they are pregnant (0 = No, 1 = Yes), if they have, or have had, a confirmed/suspected case of COVID-19 infection (0 = No, 1 = Yes), and if a close relative or friend has, or has had, a confirmed/suspected case of COVID-19 infection (0 = No, 1 = Yes). Additionally, participants were asked if they are currently, or have in the past, received mental health treatment (i.e. medication or psychotherapy) for a mental health problem (0 = No, 1 = Yes).

**Psychological indicators**. Personality traits: The Big-Five Inventory (BFI-10)[69] measures the traits of openness to experience, conscientiousness, extraversion, agreeableness, and neuroticism. Each trait is measured by two items using a five-point Likert scale that ranges from 'strongly disagree' (1) to 'strongly agree' (5). While higher scores reflect higher levels of each personality trait, and Rammstedt and John[69] reported good reliability and validity for the BFI-10 scale scores, internal reliability coefficients are not provided here. Because the scale measures each trait using only two items, coefficient alpha is inappropriate for demonstrating internal consistency[70].

*Locus of control*. The Locus of Control Scale (LoC)[71] measures internal (e.g. 'My life is determined by my own actions') and external locus of control. The latter has two components, 'Chance' (e.g. 'To a great extent, my life is controlled by accidental happenings') and 'Powerful Others' (e.g. 'Getting what I want requires pleasing those people above me'). Each subscale was measured using three questions and a seven-point Likert scale that ranges from 'strongly disagree' (1) to 'strongly agree' (7). Higher scores reflect higher levels of each construct. The internal reliabilities of the LoC subcomponents in both the Irish and UK samples were excellent (Internal $\alpha = 0.67$ & 0.71; Chance $\alpha = 0.63$ & 0.70; Powerful Other $\alpha = 0.78$ & 0.85, respectively). The internal reliabilities of the Internal and Chance subscale scores in the Irish sample were slightly lower than desirable ($\alpha = 0.67$ & 0.63, respectively) but somewhat stronger for the UK sample ($\alpha = 0.71$ & 0.70, respectively), while those for the Powerful Others subscale scores were excellent for both samples (Ireland $\alpha = 0.78$; UK $\alpha = 0.85$).

*Analytical/reflective reasoning*. The Cognitive Reflection Task (CRT)[72] is a three-item measure of analytical reasoning where respondents are asked to solve logical problems designed to hint at intuitively appealing but incorrect responses. The response format was multiple choice with three full answers (including the hinted incorrect answer), as recommended by Sirota and Juanchich[73]. The internal reliabilities of the CRT scores in the Irish and UK samples were $\alpha = 0.67$ and $\alpha = 0.69$, respectively).

*Altruism*. The Identification with all Humanity scale (IWAH)[74] is a nine-item scale adapted for use in this study (reference to 'America' in the original study was substituted with 'Ireland' or 'the UK' for this study). Respondents are asked to respond to three statements with reference to three groups – people in my community, people from Ireland/ the UK, and all humans everywhere. The three statements were presented to respondents separately for each of the three groups, as follows: (1) How much do you identify with (feel a part of, feel love toward, have concern for) …? (2) How much would you say you care (feel upset, want to help) when bad things happen to …? And, (3) When they are in need, how much do you want to help…? Response scale ranged from 1 'not at all' to 5 'very much'. Higher scores reflect greater identification with others, care for others, and a desire to help others. The internal reliabilities of each subscale of the IWAH in both the Irish and UK samples were excellent (identification with others $\alpha = 0.79$ & 0.81; care for others $\alpha = 0.88$ & 0.89; desire to help others $\alpha = 0.86$ & 0.88, respectively).

*Conspiracy beliefs*. The Conspiracy Mentality Scale (CMS)[75] measures conspiracy mindedness using five items with each scored on an 11-point scale (1 = 'Certainly not 0%' to 11 = 'Certainly 100%'). Items include, 'I think that many very important things happen in the world, which the public is never informed about', and 'I think that there are secret organisations that greatly influence political decisions'. The internal reliability of the CMS in both the Irish and UK samples was excellent ($\alpha = 0.84$ & 0.85, respectively).

*Paranoia*. The five-item persecution subscale from the Persecution and Deservedness Scale (PaDS)[76] was used. Participants rate their agreement with statements such as "I'm often suspicious of other people's intentions towards me" and "You should only trust yourself." Response options ranged from 'strongly disagree' (1) to 'strongly agree' (5) with higher scores reflecting higher levels of paranoia. The psychometric properties of the scale scores have been previously supported[77], and the internal reliability in both the Irish and UK samples was excellent ($\alpha = 0.83$ & 0.86, respectively).

*Religious and atheist beliefs*. Participants indicated their agreement to 8 statements from the Monotheist and Atheist Beliefs Scale[78]. Statements included: "God has revealed his plans for us in holy books" and "Moral judgments should be based on respect for humanity rather than religious doctrine". Response options ranged from 'strongly disagree' (1) to 'strongly agree' (5). Atheism oriented statements were reverse scored and summed with monotheist items to produce a summed score, with higher scores reflecting religious belief orientation. The psychometric properties of the scale scores have been previously supported[78], and the internal reliability in both the Irish and UK samples was excellent ($\alpha = 0.81$ & 0.83, respectively).

*Trust in institutions*. Respondents were asked to indicate the level of trust they have in political parties, the parliament, the government, the police, the legal system, scientists, and doctors and other health professionals. Responses were scored on a five-point Likert scale ranging from 'do not trust at all' (1) to 'completely trust' (5). For this study, responses to the first five institutions were summed to generate a total score for 'trust in the state'. Trust in scientists, and trust in doctors and other health professionals were treated individually.

*Authoritarianism*. The Very Short Authoritarianism Scale (VSA)[79] includes six items assessing agreement with statements such as: 'It's great that many young people today are prepared to defy authority' and 'What our country needs most is discipline, with everyone following our leaders in unity'. All items were scored on a five-point Likert scale ranging from 'strongly disagree' (1) to 'strongly agree' (5), with higher scores reflecting higher levels of authoritarianism. The internal reliability of the scale scores in the Irish sample was lower than desirable ($\alpha = 0.58$) but somewhat stronger for the UK sample ($\alpha = 0.65$).

*Social dominance*. Respondents' levels of social dominance orientation were assessed using the eight-item Social Dominance Scale (SDO7)[80]. Respondents were asked the extent to which they opposed/favoured statements such as: 'An ideal society requires some groups to be on top and others to be on the bottom'; 'Some groups of people are simply inferior to other groups'; and 'We should do what we can to equalise conditions for different groups'. Response were scored using a 5-point Likert scale ranging from 1 'Strongly oppose' to 5 'Strongly Favour'. Ho and colleagues demonstrated that the SDO7 had good criterion and construct validity[80]. The internal reliability of the scale scores in both the Irish and UK samples was excellent ($\alpha = 0.79$ & 0.82, respectively).

*Attitude towards migrants*. Two items assessing respondents' attitudes towards migrants were taken from the British Social Attitudes Survey 2015[81]. These were, (1) 'Would you say it is generally bad or good for the UK's economy that migrants come to the UK from other countries?' (scored on a 10-point scale ranging from 1 'extremely bad' to 10 'extremely good'), and (2) 'Would you say that the UK's cultural life is generally undermined or enriched by migrants coming to live here from other countries?' (scored on a 10-point scale ranging from 1 'undermined' to 10 'enriched'). These items were phrased appropriately for use with the Irish sample.

**COVID-19 information consumption and trust**. Participants were asked, 'How much information about COVID-19 have you obtained from each of these sources?', and were presented with nine options: Newspapers, Television, Radio, Internet websites, Social media, Your doctor, Other health professionals, Government agencies, and Family or friends. Responses were recorded on a four-point Likert scale (1 = none, 2 = a little, 3 = some, 4 = a lot). Participants were also asked, 'How much do you trust the information from each of these sources?', and responses were recorded on a four-point Likert scale (1 = not at all, 2 = a little, 3 = some, 4 = a lot).

**Data analysis**. The analytical strategy involved four elements linked to the study objectives. First, the proportion of people in the Irish and UK samples classified as 'vaccine accepting', 'vaccine hesitant', and 'vaccine resistant' were calculated.

Second, multinomial logistic regression analyses were performed to identify the key sociodemographic, political, and health-related indicators associated with vaccine hesitancy and resistance. Analyses were performed separately for the Irish and UK samples. For these analyses, the vaccine acceptance group was set as the reference category to identity factors associated with vaccine hesitancy and vaccine resistance, respectively. Subsequently, the models were re-estimated with the vaccine hesitant group set as the reference category to identify which factors distinguished vaccine resistant respondents from vaccine hesitant respondents. All associations between the predictor and criterion variables are represented as adjusted odds ratios (AOR) with 95% confidence intervals (i.e. all predictors are adjusted for all other covariates in each model).

Third, a series of one-way between-groups analysis of variance (ANOVA) tests with Bonferroni post-hoc tests were performed to determine on what psychological characteristics vaccine hesitant and resistant people differ from vaccine accepting people. Statistically significant differences are reported at both the standard alpha level ($p < 0.05$) and at a stricter level ($p < 0.001$) to account for the potential for increased Type 1 errors due to multiple testing. These analyses were performed for the Irish and UK samples separately. The magnitude of the effect between the three means was quantified in terms of eta-squared ($\eta^2$) where values $\leq 0.05$ reflect a small effect size, values from 0.06 to 0.13 reflect a medium effect size, and values $\geq 0.14$ reflect a large effect size[82]. Additionally, the vaccine hesitant and resistant groups were combined to represent a single group, and this group was compared to the vaccine accepting group on all psychological variables. The magnitude of the effect between the two means was quantified in terms of Cohen's d where values <0.30 reflect a small effect size, values from 0.30 to 0.80 reflect a medium effect size, and values >0.80 reflect a large effect size[78]. All data analyses were conducted using IBM SPSS Statistics for Windows Version 25.0[83]. Fieldwork and data collection was conducted by the survey company Qualtrics, which has completed more than 15,000 projects across 2500 universities worldwide. The data was generated using Qualtrics software, Version – March 2020[84].

Finally, the vaccine accepting, hesitant, and resistant groups were compared with respect to where they source their information about COVID-19, and the level of trust they have in these information sources. These comparisons were made using one-way between-groups ANOVA tests with Bonferroni post-hoc tests.

**Reporting summary**. Further information on research design is available in the Nature Research Reporting Summary linked to this article.

## Data availability

The datasets generated and/or analysed during the current study are available in the Open Science Framework (OSF) repository[85], and can be accessed here: https://osf.io/58swj/. Source data are provided with this paper.

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

## Acknowledgements
R.M. acknowledges funding support from the Cogito Foundation (Grant Nr. R10917).

## Author contributions
Conception/design of the work (J.M., P.H., F.V.); acquisition of data (all authors); analysis of data (J.M., P.H.); interpretation of findings (J.M., P.H., F.V., R.B., M.S., O.M.B., T.H., R.M.); drafted the work (J.M., P.H., F.V.); substantively revised the work (R.P.B., M.S., O.M.B., T.H., R.M., K.B., L.M., J.G.M., L.L., A.M., T.S., T.K.).

## Competing interests
The authors declare no competing interests.
