## [Peer Review File · Nature Communications]

REVIEWER COMMENTS

Reviewer #1 (Remarks to the Author):

This paper has investigated the psychological profile of vaccine hesitant/resistant individuals in the UK and Ireland. This is a well-written article, that tackles a timely issue using well-powered samples. The implications are articulated well in the discussion, where the paper will offer an important contribution to the literature. I enjoyed reading the novel paper and only have some minor comments that the author (s) may wish to consider.

The introduction provides a compelling overview of the COVID-19 landscape, and why profiling individuals who are hesitant about or resistant to a possible vaccine is important. Indeed, the aims make sense. However, to this reviewer at least, there is a lacking theoretical backing for the inclusion of the measures to identify the salient psychological characteristics. I was expecting a rationale for each measure, and thus, the justification for this being included. I was surprised to see this omitted when there is ample space to provide such a rationale. By also not including such a narrative, some key measures – such as explaining what conspiracy theories are and why they are relevant to the investigation – are missing. On a well-presented paper, this is the weakest section for me.

In the Method, a justification for the sample sizes would be welcome.

I would also encourage the reliability statistic (Cronbach's alpha) to be included in the Method, using data derived from the current study (as opposed to another published paper). Sometimes the statistics are included, and sometimes they are not. The measures are otherwise well explained.

Clarity on the grouping of the ages would be welcome (e.g., why 25-34, 35-44, etc.).

Reviewer #2 (Remarks to the Author):

The research has a very admirable objective in seeking to uncover the demographic and psychological factors that are likely to lead to acceptance/rejection of a COVID-19 vaccine. Examining where people seek information about vaccination is also a very important step in trying to improve vaccine attitudes and uptake.

I enjoyed reading the paper and it is excellent to see two nationally representative studies reported. However, my general feeling is that the findings, whilst interesting and potentially useful, are not very surprising. I therefore feel that the work might be better suited to a less high profile outlet.

To briefly explain, much of the research is descriptive – e.g., establishing just how many people are vaccine hesitant or resistant. This is good to see and national differences are interesting, but this is a very straightforward set of findings.

The other aspect of the research – looking at demographic and psychological predictors of vaccine attitudes – is of course more novel. However, I wondered how the specific psychological predictors were narrowed down. There is a large literature on vaccine hesitancy to draw upon but I cannot see any of this outline in any great detail in the introduction. The theory behind the psychological predictors is therefore lacking.

Finally, how can the authors explain differences between countries? These are two, dare I say, similar Western European countries. What would the authors expect in non-Western European countries where vaccine attitudes and vaccination rates are quite different?

Overall, I think that this is interesting research, but perhaps not suitable for this particular outlet. I do however very much appreciate the authors' research efforts at this challenging time.

Reviewer #3 (Remarks to the Author):

This is a timely paper. I only have a few comments:

1. The introduction could be shorter.
2. Line 50. Now there are a couple of effective therapeutic options available now
3. Lines 126 - 129 are not needed.
4. Please provide 95% CIs with ORs.
5. Line 156 and other places. These re not psychological indicators; these are attitudes.
6. Lines 230-232 are confusing.
7. 293 -295: how?

Reply to reviewer comments

We would like to sincerely thank the anonymous reviewers for their careful reading of our manuscript. We have gratefully received these comments and have revised our manuscript accordingly. We believe the changes have significantly improved our paper. Before responding to each reviewer on a point-by-point basis, we felt that it would be helpful to provide a general overview of the major revisions that we have made to our manuscript since many of these edits are pertinent to comments made by multiple reviewers. To make identifying revisions easier, we have highlighted all changes to the manuscript in yellow.

Overview of Major Revisions

First, Reviewers 1 and 2 requested that we provide a justification for the selection of the psychological variables used in this study. While noting Reviewer 3's request to reduce the length of the introduction, we agree that it was important to provide a theoretical and empirical rationale for the selection of all psychological variables used in our analyses. We appreciate the opportunity to do this as it should be extremely helpful for readers who may be unfamiliar with the psychological literature underpinning our work. In the revised manuscript, we feel that we have struck an appropriate balance in giving sufficient explanation for why each psychological variable is relevant, while also maintaining a succinct and focused introduction.

Second, after we submitted our manuscript a number of new developments have occurred that are related to our study, including progress on a COVID-19 vaccine and published data regarding attitudes towards a potential COVID-19 vaccine. We have revised our manuscript accordingly to acknowledge these important developments.

Finally, Reviewer 1 noted that internal reliability statistics were missing for some measures used in our study. In reporting these missing statistics, we discovered that the Locus of Control measure had not been scaled correctly in both datasets. While this issue did not affect any of the substantive findings we originally presented, it did require us to update our estimates presented in Tables 3 and 4 (highlighted in yellow).

We would like also to take the opportunity here to repeat our response to the (since rescinded) request we received when invited to revise and resubmit our manuscript – i.e. to supplement our findings using data from a non-Western nation. This response read:

*“First, we are not in possession of such data, nor do we know of any available data that contains the measures we have analysed in our multi-country study. Given that the UK is one of the world's most affected countries, (third highest COVID-19 related death toll), we believe our results for this country alone are extremely important. Second, if we were to endeavour to secure such data, the interpretability/utility of our current set of findings would be compromised as the data collection timeframes and contexts would be inconsistent. Third, and most importantly, time is of the essence for this particular set of findings. While we recognise and agree that a broader cultural comparison would undoubtedly enhance the value of the paper (in fact, all published research could be improved by analysing additional data), **we believe that the key aim of our study – a call to***

arms to help policymakers and health officials increase vaccine uptake – has the potential not only to stimulate much needed debate, discussion and planning relating to future COVID-19 inoculation, but to encourage the important international data collection, analysis and comparisons that you are requesting.”

We believe that our last statement in this reply (in bold) highlights that which is most salient and valuable about this particular study and that all countries should be motivated to conduct similar research to prepare their respective populations for a future COVID-19 vaccine. The ‘who’ and ‘where’ of this study therefore are not as important as the ‘why’ and the ‘when’. While international and cross-cultural comparisons will undoubtedly be of future value, assessing, identifying and understanding why people may be vaccine hesitant (to a COVID-19 vaccine specifically), before a vaccine has been successfully delivered, surely is of unquestionable value to all those who must prepare for the distribution and uptake of a future vaccine. We hope you can support us to share this work and these findings with the widest possible audience.

Below, we provide detailed responses to each of the reviewers.

Reviewer #1 (R1):

R1: “This paper has investigated the psychological profile of vaccine hesitant/resistant individuals in the UK and Ireland. This is a well-written article, that tackles a timely issue using well-powered samples. The implications are articulated well in the discussion, where the paper will offer an important contribution to the literature. I enjoyed reading the novel paper and only have some minor comments that the author (s) may wish to consider.”

Authors’ response - Thank you for your positive comments.

R1: “The introduction provides a compelling overview of the COVID-19 landscape, and why profiling individuals who are hesitant about or resistant to a possible vaccine is important. Indeed, the aims make sense. However, to this reviewer at least, there is a lacking theoretical backing for the inclusion of the measures to identify the salient psychological characteristics. I was expecting a rationale for each measure, and thus, the justification for this being included. I was surprised to see this omitted when there is ample space to provide such a rationale. By also not including such a narrative, some key measures – such as explaining what conspiracy theories are and why they are relevant to the investigation – are missing. On a well-presented paper, this is the weakest section for me.”

Authors’ response - We had refrained from including this level of detail in our original manuscript to produce a short and focussed rationale relating to the study objectives. We were concerned that inclusion of a justification for each psychological variable would result in an introduction that was unacceptably long. We note that Reviewer 2 also asked for the inclusion of such a rationale while Reviewer 3 suggested that our introduction was overly long. Given the balance of reviewer comments, we decided to include two new paragraphs to the introduction (pages 4 & 5) to provide an empirical rationale for the selection of our psychological variables, and a theoretical rationale for their importance in understanding of hesitance/resistance to a vaccine for COVID-19. We hope that the manner in which we have presented these paragraphs strikes a reasonable balance between the competing requests of our reviewers and a desire to succinctly frame our study objectives.

R1: “In the Method, a justification for the sample sizes would be welcome.”

Authors’ response - We have now included a clear and detailed justification for each of our survey sample sizes. The new text (page 19) reads: “Power analyses were conducted to determine the optimal sample sizes for both countries. As the C19PRC Study was primarily concerned with tracking mental health disorders (depression, generalized anxiety disorder [GAD], and posttraumatic stress disorder [PTSD]) in the general population, sample size calculations were based on existing prevalence estimates for these disorders. In Ireland and the UK, the estimated prevalence of PTSD is 5% and 4%, respectively, and lower than the prevalence estimates of depression and generalized anxiety.⁶⁷ To detect a disorder with a prevalence of 4%, with precision of 1%, and 95% confidence level, a sample size of 1476 was required. The survey company used to collect the data could only guarantee a maximum

sample size of 1,000 participants in Ireland, whereas a larger sample could be obtained in the UK. This is a consequence of the much smaller population of Ireland (4.9 million people) compared to the UK (66.7 million people). Therefore, the target sample size in Ireland was set at 1,000 which, holding all other parameters in the sample size calculation equal, resulted in a precision of 1.21%. In the UK, a target sample was set at 2,000 to increase the number of 'cases' detected because of the intention to track changes in the mental health problems in the population over time."

R1: "I would also encourage the reliability statistic (Cronbach's alpha) to be included in the Method, using data derived from the current study (as opposed to another published paper). Sometimes the statistics are included, and sometimes they are not. The measures are otherwise well explained."

Authors' response – We apologise for this oversight in the initial submission. Internal reliability estimates for all scale scores in the Irish and UK samples are now included. However, we do not report internal reliability estimates for the scale scores measuring the five personality traits. This is due to the fact that The Big-Five Inventory measures each trait using only two items, and it is well documented that coefficient alpha is inappropriate and meaningless for two-item scales (Eisinga, Grotenhuis, & Pelzer, 2013). We have included a footnote in the manuscript to explain this also.

Eisinga R, Grotenhuis Mt, Pelzer B. The reliability of a two-item scale: Pearson, Cronbach, or Spearman-Brown?. *Int J Public Health*. 2013;58(4):637-642. doi:10.1007/s00038-012-0416-3

R1: "Clarity on the grouping of the ages would be welcome (e.g., why 25-34, 35-44, etc.)."

Authors' response - The International Journal of Epidemiology provides guidelines on forming age categories: "grouping should be mid-decade to mid-decade or in five-year age groups (e.g. 35–44 or 35–39, 40–44, etc, but not 20–29, 30–39 or other groupings).

Reijneveld SA Age in epidemiological analysis *Journal of Epidemiology & Community Health* 2003;57:397.

We have now included text detailing this in the method section. This reads: "In compliance with the International Journal of Epidemiology guidelines on forming age categories we categorised age from mid-decade to mid-decade (e.g. 35–44, 45–54).⁶⁸"

Reviewer #2 (R2):

R2: “The research has a very admirable objective in seeking to uncover the demographic and psychological factors that are likely to lead to acceptance/rejection of a COVID-19 vaccine. Examining where people seek information about vaccination is also a very important step in trying to improve vaccine attitudes and uptake.”

Authors’ response - Thank you. We believe this to be a valuable contribution to the research community while vaccine trials are underway.

R2: “I enjoyed reading the paper and it is excellent to see two nationally representative studies reported. However, my general feeling is that the findings, whilst interesting and potentially useful, are not very surprising. I therefore feel that the work might be better suited to a less high-profile outlet. To briefly explain, much of the research is descriptive – e.g., establishing just how many people are vaccine hesitant or resistant. This is good to see and national differences are interesting, but this is a very straightforward set of findings.”

Authors’ response – We appreciate that some of our initial findings are indeed descriptive in nature. However, our findings have considerable importance to the global public health effort to combat COVID-19, and to be fair, we provide much more than a simple descriptive account later in our manuscript. For context: Since the submission of our manuscript, an editorial has been published in *The European Journal of Health Economics* that provides an estimate of the proportion of the adult populations across seven European countries who are accepting of, hesitant about, and resistant to a vaccine for COVID-19 (Neumann-Böhme et al., 2020). These findings align with our own findings that about one-quarter of adults in the general population are hesitant or resistant to COVID-19 vaccination. Yet, the more important questions, are (1) *why* are so many people hesitant or resistant to such a vaccine, and (2) *what* can be done to address this problem?

Thus, we have attempted to push scientific understanding beyond descriptive analyses and instead into *explaining* why we observe these patterns in the data. In our manuscript we discuss the above questions in terms of (a) *identifying* the problem, (b) *understanding* its root causes, and (c) *addressing* the problem. We therefore believe that our work is crucially important and relevant because it provides: (a) a clear description of the sociodemographic profile of those in society who are hesitant and resistant to a COVID-19 vaccine; (b) a clear psychological profile of these persons; and (c) practical suggestions for how public health officials can use this information to most effectively target and reach those who are vaccine hesitant and resistant. We present this information as clearly as possible in the discussion section so that these findings are can be taken up by public health authorities around the world (see ‘*The psychological profile of COVID-19 vaccine hesitant and resistant people*’ section on page 14 for an explicit profile summary that has the potential to directly inform those responsible for public health messaging). Thus, we believe that this work is indeed well-suited to publication in *Nature Communications*, given the journal’s global reach.

Neumann-Böhme, S., Varghese, N.E., Sabat, I. *et al.* Once we have it, will we use it? A European survey on willingness to be vaccinated against COVID-19. *Eur J Health Econ* **21**, 977–982 (2020). <https://doi.org/10.1007/s10198-020-01208-6>

R2: “The other aspect of the research – looking at demographic and psychological predictors of vaccine attitudes – is of course more novel. However, I wondered how the specific psychological predictors were narrowed down. There is a large literature on vaccine hesitancy to draw upon but I cannot see any of this outline in any great detail in the introduction. The theory behind the psychological predictors is therefore lacking.”

Authors’ response – We agree that this is important information and have thus revised the introduction to include a justification for the selection of all the psychological variables used in our analyses. More detail about these revisions is provided in the *Overview of Revisions* and our response to R1 above.

R2: “Finally, how can the authors explain differences between countries? These are two, dare I say, similar Western European countries. What would the authors expect in non-Western European countries where vaccine attitudes and vaccination rates are quite different?”

Authors’ response – We acknowledge R2’s concerns that the UK and Ireland may be too similar given their geographic proximity; however, we would say that these countries differ significantly in historical, cultural, and political terms, and most importantly to this study, in *how they have responded to the COVID-19 pandemic*. To set aside the historical and cultural differences between the nations, while both countries are located in Western Europe, the Republic of Ireland is very strongly ‘pro-EU’ whereas the UK has voted to leave the EU. Ireland is also ranked in global terms as a far more politically stable nation than the UK (24th versus 96th in a global list: see https://www.theglobaleconomy.com/rankings/wb_political_stability/).

Regarding the response to the COVID-19 pandemic, the Republic of Ireland responded much more quickly to the outbreak of COVID-19 than did the UK. As we have alluded to in the method section, the government of the Republic of Ireland initiated lockdown measures very soon after the first confirmed case of COVID-19 on the island of Ireland (the first known case was actually confirmed in Belfast, Northern Ireland). In stark contrast, the UK government was much slower to initiate quarantine measures and, for example, was still permitting large scale public gatherings to occur when Ireland had closed schools and businesses. As a result, the death rate from COVID-19 has been significantly lower in Ireland than in the UK: the latter’s death rate from COVID-19 is more than twice as high as in Ireland (272.21 per 1 million vs. 123.5 per 1 million) (see <https://ourworldindata.org/grapher/total-covid-deaths-per-million?year=2020-04-20>). Additionally, citizens of Ireland have expressed very high levels of satisfaction in how their government has handled the pandemic (87% - 5th highest of 53 countries surveyed), whereas the proportion of the UK citizens that expressed satisfaction in how the

government handled the crisis was far lower (58% - 41st of 53 countries surveyed) (see <https://daliaresearch.com/blog/democracy-perception-index-2020/>).

All this is to say, that while the UK and Ireland may be geographically close, they are actually very different in ways that matter to the research questions at hand in this paper. Indeed, this point is made even clearer by the different patterns of sociodemographic variables associated with vaccine hesitant/resistant attitudes in each country.

To make this point more salient, we have added a couple of sentences to the discussion where we discuss the context-dependent nature of the sociodemographic predictors of COVID-19 vaccine hesitance and resistance. This reads: "Similarity in sociodemographic predictors of COVID-19 vaccine hesitance and resistance across Ireland and the UK may not be considered unusual given their geographical proximity; however, the two nations are quite distinct historically, politically, and culturally, and in their response to the COVID-19 pandemic. To the latter point, the UK has a death rate from COVID-19 that is more than twice that of Ireland (272.21 per 1 million vs. 123.5 per 1 million).⁴² Thus, it was unsurprising that context-dependent predictors of vaccine hesitance/resistance emerged."

Moreover, we have expanded the discussion in places to recognise the need for the type of research we have conducted to be extended to non-Western nations. Additionally, given the context-dependent nature of which demographic factors predict vaccine hesitant/resistant attitudes, we have added points to the discussion relating to the need for public health authorities to replicate our work in their respective nations so as to identify groups that are particularly high-risk. We make the point that only a global vaccination programme will be truly successful in suppressing the virus and, as per the WHO's SAGE group on Immunization advice, identifying anti-vaccine hotspots and pre-emptively targeting these groups is an essential public health requirement.

R2: "Overall, I think that this is interesting research, but perhaps not suitable for this particular outlet. I do however very much appreciate the authors' research efforts at this challenging time."

Authors' response – We respectfully disagree and believe that our findings are of sufficient global public health importance to warrant publication in such a high-profile outlet. Nature Communications has a global reach and a weight of impact that increases the likelihood that public health officials around the world will encounter this work and reflect upon its implications. In addition, we hope our work will draw attention to the importance of collecting psychologically-rich data, which can then be used to inform public health messaging and policies aimed at increasing vaccine uptake. Thus, we believe our manuscript provides an important contribution at such a crucial point in time to the preparatory work needed for wide-scale delivery of a vaccine against COVID-19.

Reviewer #3 (R3):

R3: "This is a timely paper."

Authors' response - Thank you for recognising the timeliness of our research.

R3: "The introduction could be shorter."

Authors' response - Brevity was our original intention for the introduction; however, as R1 and R2 have requested justifications for the selection of the psychological variables included in our analyses, we have added slightly to the introduction to satisfy these requests. We have made every effort to present the background, rationale, and study objectives as succinctly as possible, and we hope R3 understands our reason for adding material to the introduction in this revision.

R3: "Line 50. Now there are a couple of effective therapeutic options available now."

Authors' response – This comment is well taken, however, in this sentence we wanted to convey the message that, upon the outbreak of COVID-19, governments around the world were forced to initiate widescale quarantine measures because of the absence of any effective therapeutics or a vaccine *at the time*. We have updated the relevant opening paragraphs of the introduction to note the changes in COVID-19 cases and deaths, as well as the progress toward the development of a vaccine. We now note that there are currently 8 vaccines in Phase 3 clinical trials and 2 are approved for early or limited use.

R3: "Lines 126 - 129 are not needed."

Authors' response – We agree. These lines have been deleted from the manuscript.

R3: "Please provide 95% CIs with ORs."

Authors' response - 95% CIs now accompany all AORs

R3: "Line 156 and other places. These re not psychological indicators; these are attitudes."

Authors' response - We respectfully disagree with the reviewer here. These indicators are widely recognised and studied psychological constructs measured using validated psychometric scales.

R3: "Lines 230-232 are confusing."

Authors' response - This has been amended and now reads: "Thus, upwards of approximately 10% of study populations appear to be opposed to vaccinations in whatever form they take."

R3: “293 -295: how?”

Authors’ response - We have offered clarification here. The discussion has been expanded and, in full, now states: “Responsibility for public health messaging primarily lies with governments, scientists, and medical professionals. However, our results suggest that health messaging delivered by these groups may not only be ineffective for vaccine hesitant or resistant individuals, but may also account for ‘backfire effects’ (i.e., the tendency for people to hold to a belief more strongly when presented with evidence against that belief) observed elsewhere in the vaccine literature.⁵⁹ The high level of distrust that vaccine hesitant and resistant people have for those who represent established authority is likely to provoke psychological resistance to any message emanating from these sources, and to an entrenchment of their existing ‘anti-establishment’ or ‘anti-authority’ beliefs. Consequently, anti-vaccine beliefs may be expressed by some individuals in society as a way to advertise their ‘anti-establishment’ sentiments. By understanding the psychological dispositions of these individuals, another – potentially more effective – approach could be adopted. For example, recognising their preference for social dominance and authoritarianism, and their distrust of conventional authority figures, vaccine hesitant or resistant persons may be more receptive to authoritative messages regarding COVID-19 vaccine safety and efficacy if it is delivered by individuals within non-traditional positions of authority and expertise. Engagement of religious leaders, for example, has been documented as an important approach to improve vaccine acceptance among some groups.⁶⁰ In the specific context of Ireland, results further suggest that independent politicians and politicians from the anti-establishment Sinn Féin party may represent key figures to convey the importance of vaccination against COVID-19 to many in society. Key to the preparation of a COVID-19 vaccine is, therefore, the early and frequent engagement of religious and community-leaders,⁶¹ and for health authorities to work collaboratively with multiple societal stakeholders to avoid the feeling that they are only acting *on behalf* of government authorities.⁶²”

REVIEWERS' COMMENTS

Reviewer #1 (Remarks to the Author):

Thank you for addressing my comments. I do believe that this is a much stronger manuscript.

Reviewer #3 (Remarks to the Author):

I find the responses to my comments satisfactory.

I also agree with the authors' responses regarding the suitability of this paper in a "high profile" journal.

Reply to reviewer comments

We would like to sincerely thank the anonymous reviewers for their careful reading of our manuscript. We have gratefully received these comments and have revised our manuscript accordingly. We believe the changes have significantly improved our paper.

REVIEWERS' COMMENTS

Reviewer #1 (Remarks to the Author):

Thank you for addressing my comments. I do believe that this is a much stronger manuscript.

Authors' response - Thank you for your positive comments.

Reviewer #3 (Remarks to the Author):

I find the responses to my comments satisfactory.

I also agree with the authors' responses regarding the suitability of this paper in a "high profile" journal.

Authors' response - Thank you for your positive comments.